# Tropical cyclone exposure is associated with increased hospitalization rates in older adults

Robbie M. Parks [1,2 ✉], G. Brooke Anderson[3], Rachel C. Nethery[4], Ana Navas-Acien[2], Francesca Dominici[4] & Marianthi-Anna Kioumourtzoglou[2]

Hurricanes and other tropical cyclones have devastating effects on society. Previous case studies have quantified their impact on some health outcomes for particular tropical cyclones, but a comprehensive assessment over longer periods is currently missing. Here, we used data on 70 million Medicare hospitalizations and tropical cyclone exposures over 16 years (1999–2014). We formulated a conditional quasi-Poisson model to examine how tropical cyclone exposure (days greater than Beaufort scale gale-force wind speed; ≥34 knots) affect hospitalizations for 13 mutually-exclusive, clinically-meaningful causes. We found that tropical cyclone exposure was associated with average increases in hospitalizations from several causes over the week following exposure, including respiratory diseases (14.2%; 95% confidence interval [CI]: 10.9–17.9%); infectious and parasitic diseases (4.3%; 95%CI: 1.2–8.1%); and injuries (8.7%; 95%CI: 6.0–11.8%). Average decadal tropical cyclone exposure in all impacted counties would be associated with an estimated 16,772 (95%CI: 8,265–25,278) additional hospitalizations. Our findings demonstrate the need for targeted preparedness strategies for hospital personnel before, during, and after tropical cyclones.

[1] The Earth Institute, Columbia University, New York, NY, USA. [2] Department of Environmental Health Sciences, Mailman School of Public Health, Columbia University, New York, NY, USA. [3] Department of Environmental & Radiological Health Sciences, Colorado State University, Fort Collins, CO, USA. [4] Department of Biostatistics, T. H. Chan School of Public Health, Harvard University, Boston, MA, USA. ✉email: robbie.parks@columbia.edu

Tropical cyclones, such as hurricanes and tropical storms, have a devastating impact on the economy[1–3], environment[4,5], and human health[6–8]. Exposure to such events is an important public health concern and one of the key drivers for seeking disaster risk reduction measures[9]. The intensity of tropical cyclones is predicted to change due to anthropogenic climate change[10–19]. Land subsidence[20] and increases in the proportion of impervious surfaces[21] may further exacerbate cyclone impacts. A recent report by the National Centers for Environmental Information estimated that storm-related costs in the United States between 1980 and 2017 exceeded $1.3 trillion[22]. Previous assessments of health impacts have largely focused on single extreme events, including well-known examples, such as Hurricane Katrina in New Orleans (2005)[23–25] and Hurricane Sandy in New York City (2012)[4,26,27].

Some studies have reviewed general evidence of health impacts of storms and hurricanes, primarily using case studies, for cardiovascular diseases, respiratory diseases, dialysis-, and injury-related hospitalizations, showing harmful impacts overall[6,28–33]. Other studies have previously used claims and Medicare data to measure impacts of disasters, including how mortality and morbidity was affected in the Medicare population after Hurricane Katrina[34], and changes in Medicare cost and utilization[35]. Findings from these studies motivate exploring whether other causes of hospitalization are impacted by tropical cyclones. There are plausible behavioral and physiological pathways for a relationship between tropical cyclone exposure and many adverse health outcomes—such as respiratory complications, due to electricity cuts affecting breathing apparatus[6,36], injuries from trying to evacuate or repair a damaged property[6,26], or dietary problems due to disrupted food supply lines[6,37]. Despite these prior findings and biological plausibility, there is an overall knowledge gap in consistently and comprehensively quantifying how tropical cyclone exposure drives hospitalizations across time and space.

In this work, our aim was to evaluate how hospitalizations from various causes in the United States are associated with tropical cyclone exposure that occurs today, and could become increasingly intense, on average, as a result of global climate change[13–15]. We found that (i) tropical cyclone exposure was associated with overall increases in hospitalization rates for many causes and sub-causes in the week after exposure, with decreases for some chronic conditions; (ii) hurricane-force tropical cyclone exposure amplified the impact of weaker winds on hospitalization rates; and (iii) after tropical cyclone exposure, increases in hospitalization rates were driven by increases in emergency hospitalizations, with decreases in rates driven by decreases in nonemergency hospitalizations.

## Results

**Tropical cyclone exposure**. We assigned tropical cyclone exposure to a particular day and county if the peak sustained wind that day in that county reached or exceeded gale force (≥34 knots) on the Beaufort scale, when the tropical cyclone was at the point of closest approach to that county, described in "Methods". In total, there were 2547 tropical cyclone exposure days in 898 counties during our study period (1999–2014). By county, the number of tropical cyclone exposure days across our study period ranged from 1 to 15, with median of 2 and mean of 2.8. Tropical cyclone exposure occurred from May to October, with the greatest occurrence in September ($n = 1337$ tropical cyclone exposure days; 52% of all tropical cyclone exposure days). Tropical cyclone exposures were most frequent in the eastern and south-eastern coastal counties (Fig. 1). North Carolina was the single state with the most days of tropical cyclone exposure during the period ($n = 413$), with Jones and Pamlico Counties, both in North Carolina, each experiencing the most county-level exposure days ($n = 15$).

**Medicare hospitalizations**. We used data on enrollees from the dynamic Medicare cohort in the 898 counties from 30 states and districts in the United States, which experienced at least one tropical cyclone during our 16-year study period, with information on underlying primary cause of hospitalization and county of residence. From 1999 to 2014, there were 69,682,674 Medicare hospitalizations in the 898 counties impacted by tropical cyclones (Supplementary Table 1). The hospitalizations from these counties included 47.2% of all hospitalizations nationwide during our study period. We grouped the 15,072 possible International Classification of Diseases, Ninth Revision, Clinical Modification (ICD-9-CM) codes into 13 main causes (Fig. 2), using the Clinical Classifications Software (CCS) algorithm (Supplementary Table 2)[38]. Hospitalizations from these 13 causes accounted for 94.8% of the total hospitalizations in our study. Hospitalizations not included in these causes were other hospitalizations (Fig. 2), mainly including rare compilations during pregnancy and ill-defined causes. Cardiovascular (30%), respiratory (12.5%), and digestive system diseases (10%) were the three leading hospitalization causes in our study (Supplementary Table 1).

**Association of tropical cyclones exposure with hospitalization rates**. We analyzed the association between tropical cyclone exposure and daily hospitalization rates up to 7 days after the day of exposure, using a conditional quasi-Poisson model, described in detail in "Methods". We present these results in Fig. 3, which displays results as relative (percentage) changes in hospitalization rates after tropical cyclone exposure. We observed the highest increases in hospitalization rates from respiratory diseases; increases occurred across all studied days after the day of exposure, peaking 1 day after exposure (23.8%; 95% confidence interval [CI]: 18.6, 29.3%). Injury hospitalization rates increased across all studied days after the day of exposure, with a peak at 2 days after the day of exposure (13.5%; 95% CI: 8.5, 18.9%). We observed decreased hospitalization rates on the day of exposure for all other causes. For several causes (cardiovascular diseases, endocrine disorders, genitourinary diseases, infectious and parasitic diseases, nervous system diseases, and skin and subcutaneous tissue diseases) hospitalization risk followed a similar pattern, decreasing on the day of exposure, peaking 1–3 days later, and gradually returning to the rate expected during unexposed days within about a week.

In Fig. 4, we present average relative (percentage) changes in hospitalization rates across the eight examined lag days across the 13 causes in the main analysis, as well as for sub-causes with at least 50,000 hospitalizations during our study period. The sub-causes are linked to the 13 main causes in Supplementary Table 2. Respiratory diseases exhibited the largest average increase in hospitalizations (14.2%; 95% CI: 10.9, 17.9%). We observed the largest decreases in hospitalization rates for cancers (−4.4%; 95% CI: −2.9, −5.8%). Cardiovascular diseases did not change overall (−0.3%; 95% CI: −1.3, 0.7%). There was variation in changes to hospitalization rates for hospitalization sub-causes within each larger cause. For example, within respiratory diseases, we observed the largest increase in hospitalization rates for chronic obstructive pulmonary disease (COPD; 44.7%; 95% CI: 36.6, 54.2%) and the largest decrease in other upper respiratory infections (−8.8%; 95% CI: −6.8, −10.1%).

In addition, we examined the distinct impact of tropical cyclone exposures, in which the county's peak sustained wind was hurricane force (Beaufort scale hurricane-force winds, ≥64 knots) compared to tropical cyclone exposures with lower local winds

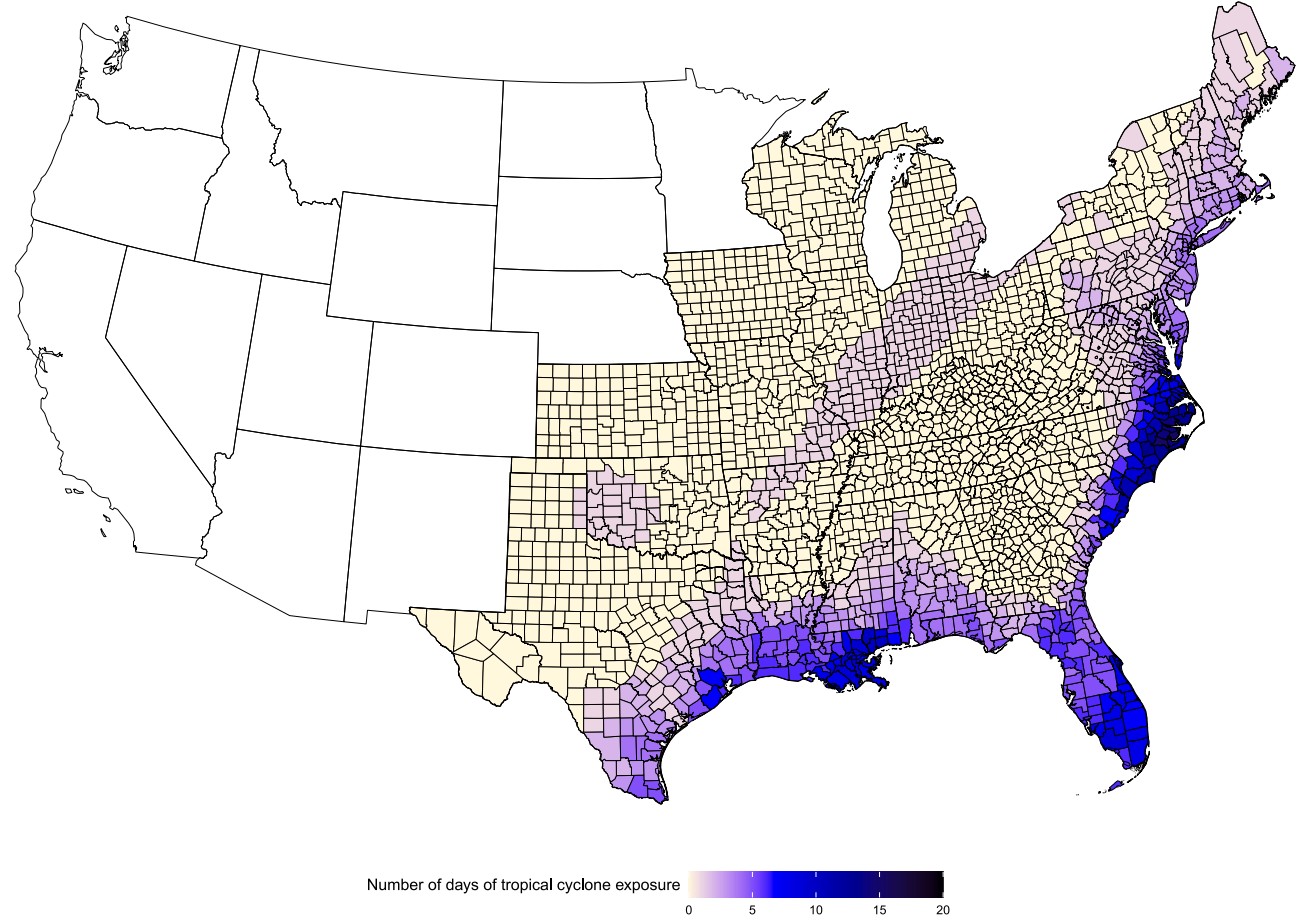

**Fig. 1 Tropical cyclone exposure days.** Number of days with tropical cyclone exposure by county for 1999–2014.

(Beaufort scale gale- to violent storm-force winds, ≥34 to <64 knots; Fig. 5). Of the 2547 county-day exposures (Fig. 1), 116 (5%) were hurricane force and came from 20 hurricanes. Across causes, the relative (percentage) changes in hospitalization rates during hurricane-force exposures broadly amplified the overall tropical cyclone effects presented in Fig. 3. We observed the highest estimates for respiratory disease hospitalizations, with an 100.3% (95% CI: 77.2, 126.4%) increase 1 day after hurricane exposure compared with a 17.1% (95% CI: 11.8, 22.7%) increase 1 day after exposure to tropical cyclones with lower local winds.

We also examined the association between tropical cyclone exposure and daily hospitalization rates by type of hospital admission (emergency vs. nonemergency; Fig. 6). Generally, nonemergency hospitalization rates decreased in the first few days after tropical cyclone exposure before returning to no change in subsequent days, with the exception of infectious and parasitic diseases, for which we estimated increases in nonemergency hospitalizations for lags 1, 2, and 4. Emergency hospitalization rates for cardiovascular diseases, respiratory diseases, and injuries increased in the days after tropical cyclone exposure, with other causes generally showing lower or no decreases across days, in comparison to nonemergency hospitalizations.

**Additional hospitalizations for tropical cyclone exposure.** Finally, we estimated the total number of additional hospitalizations for tropical cyclone exposure per decade in the week following the day of exposure across all counties included in our analysis. We used the relative (percentage) changes in hospitalization rate estimates of each cause on day of and each day after exposure, as shown in Figs. 3 and 4, along with the average

hospitalization rates during May to October in 1999–2014 and average decadal tropical cyclone exposure, described in detail in "Methods". Based on this analysis, there would be an estimated 16,772 (95% CI: 8265, 25,278) excess hospitalizations per decade in the eight days (i.e., day of and up to 7 days after exposure), following tropical cyclone exposures in the 30 states and districts included in our analysis (Fig. 7). Respiratory diseases would make up the largest number of additional hospitalizations with 16,413 (95% CI: 13,054, 19,925), followed by injuries with 10,645 (95% CI: 7767, 13,660) and infectious and parasitic diseases with 2185 (95% CI: 458, 4,041). Musculoskeletal and connective tissue disease-related hospitalizations would decrease the most, with 5899 (95% CI: 3970, 7723) fewer overall hospitalizations.

**Sensitivity analyses.** Our results were robust to sensitivity analyses of several methods for temperature adjustment. We also fit models (1) including the temperature on the day of tropical cyclone exposure, as well as temperature terms of up to 7 days after exposure and (2) without a temperature term. Our results were robust to these sensitivity analyses.

## Discussion
We used data over 16 years on 70 million Medicare hospitalizations and a comprehensive database of county-level local winds associated with tropical cyclones to examine how tropical cyclone wind exposures affect hospitalizations from 13 mutually-exclusive, clinically-meaningful causes, along with common sub-causes. We found the highest increases in respiratory disease hospitalization rates the day of and up to 7 days after the day of tropical cyclone exposure. Hospitalization rates from several

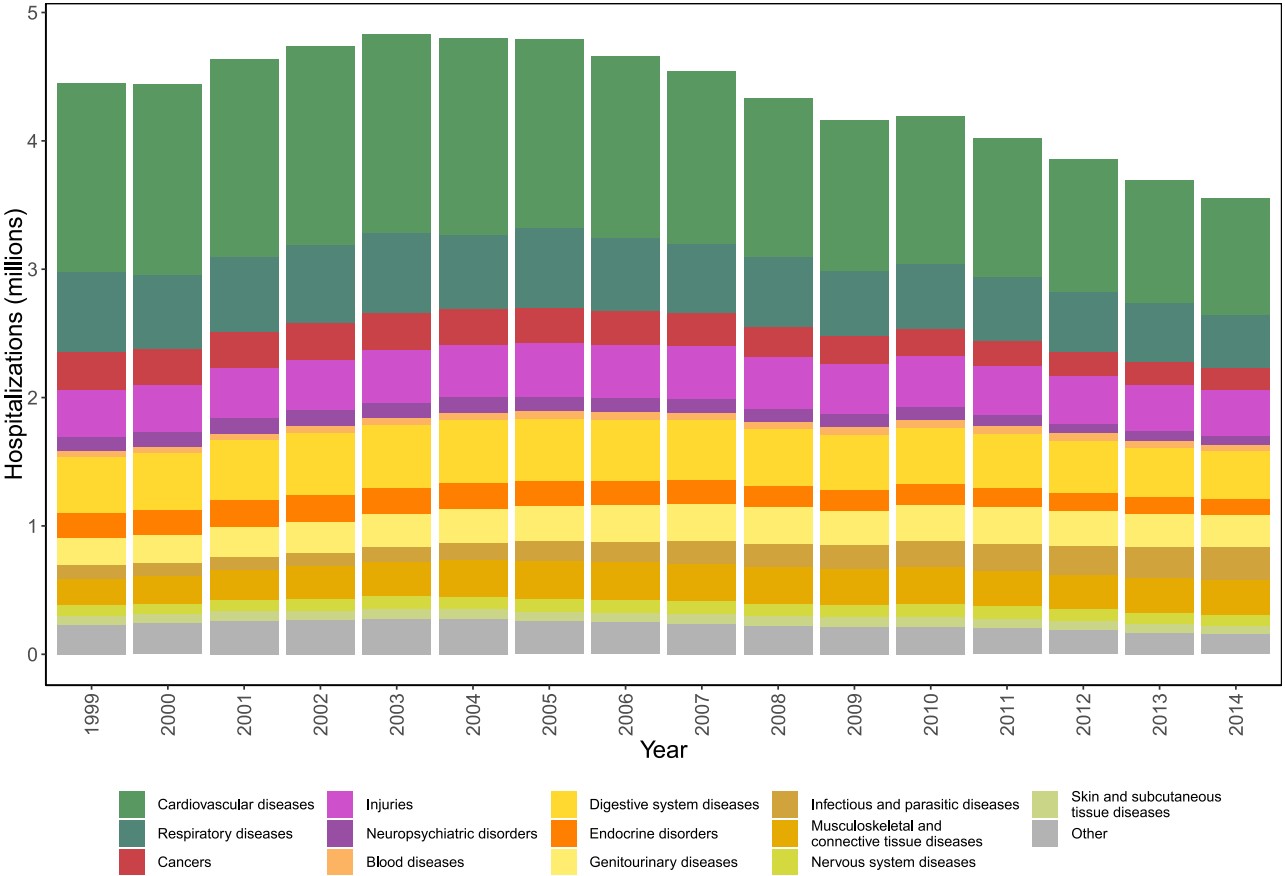

**Fig. 2 Annual Medicare hospitalizations by cause.** Number of Medicare hospitalizations by year and cause of hospitalization for counties with at least one tropical cyclone exposure for 1999–2014.

causes decreased on the day of exposure, then increased 1–3 days after the day of exposure, returning back to no association by 7 days after. There was variation by sub-cause of hospitalization within each of the 13 causes. Hurricane-force tropical cyclone exposure amplified the impacts compared to cyclones with weaker winds. Changes in emergency hospitalization rates drove increases in hospitalization rates, with decreases driven by reductions in nonemergency hospitalizations. By our analysis, on average there would be increases in overall number of hospitalizations following tropical cyclone wind exposure, with the largest increases in respiratory diseases and injuries.

**Acute vs. chronic causes of hospitalization.** The observed decreases in hospitalization rates on the day of exposure for most causes are plausible, as the immediate risk to life by making a journey during a tropical cyclone may deter many from seeking treatment at hospital. In general, the decision to make an urgent visit to a hospital for the treatment would be largely based on patients' feeling of pain, shortness of breath, and which symptoms of a potential adverse health condition they may have noticed. Unless a negative health outcome is so acute that it requires immediate treatment, patients may delay care due to risk of additional harm from tropical cyclone exposure. Consistent with this, we observed general increases in average hospitalization rates in a county exposed to tropical cyclone winds for more acute adverse health outcomes, such as COPD and leukemia, while hospitalizations for chronic conditions, such as hemorrhoids or osteoarthritis, decreased. Canceled inpatient appointments might also play a key factor here, with nonemergency procedures being delayed or rescheduled[39]. The subsequent peak 1–3 days after

exposure may in part be driven by patients visiting the hospital for care missed at locations other than the hospital (e.g., at home or at the family physician's offices) due to disruption from a tropical cyclone. There is also evidence that proximity to a tropical cyclone's path may result in the area's ambulatory (outpatient) care being disrupted[39].

**Direct vs. indirect tropical cyclone hospitalizations.** Tropical cyclone wind exposure can impact hospitalizations via direct (e.g., from physical trauma during exposure) or indirect (e.g., disrupting normal care management at local health care providers, causing damage to critical infrastructure which subsequently impacts health, or via longer-term impacts from stress) pathways[6]. Direct impacts would have more immediate impacts on hospitalizations. Longer-term, indirect impacts, such as from stress to do with the loss of property from a disaster, may not be fully measured by our analysis, as they could manifest themselves further in time than a week after a disaster[40]. There may be cases where tropical cyclone exposure could prevent normal medical care or management, compelling people go to the hospital to access resources that they would otherwise get outside the hospital without the storm.

**Respiratory diseases.** One likely explanation for the elevated rates in respiratory hospitalizations following tropical cyclone exposure is that those with respiratory issues may need power for medical equipment to breathe[6]. Power outages commonly result from tropical cyclone winds[36]. During or following a tropical cyclone, the potential loss of power can trigger a faster hospital visit for some in this group, as existing chronic conditions may become

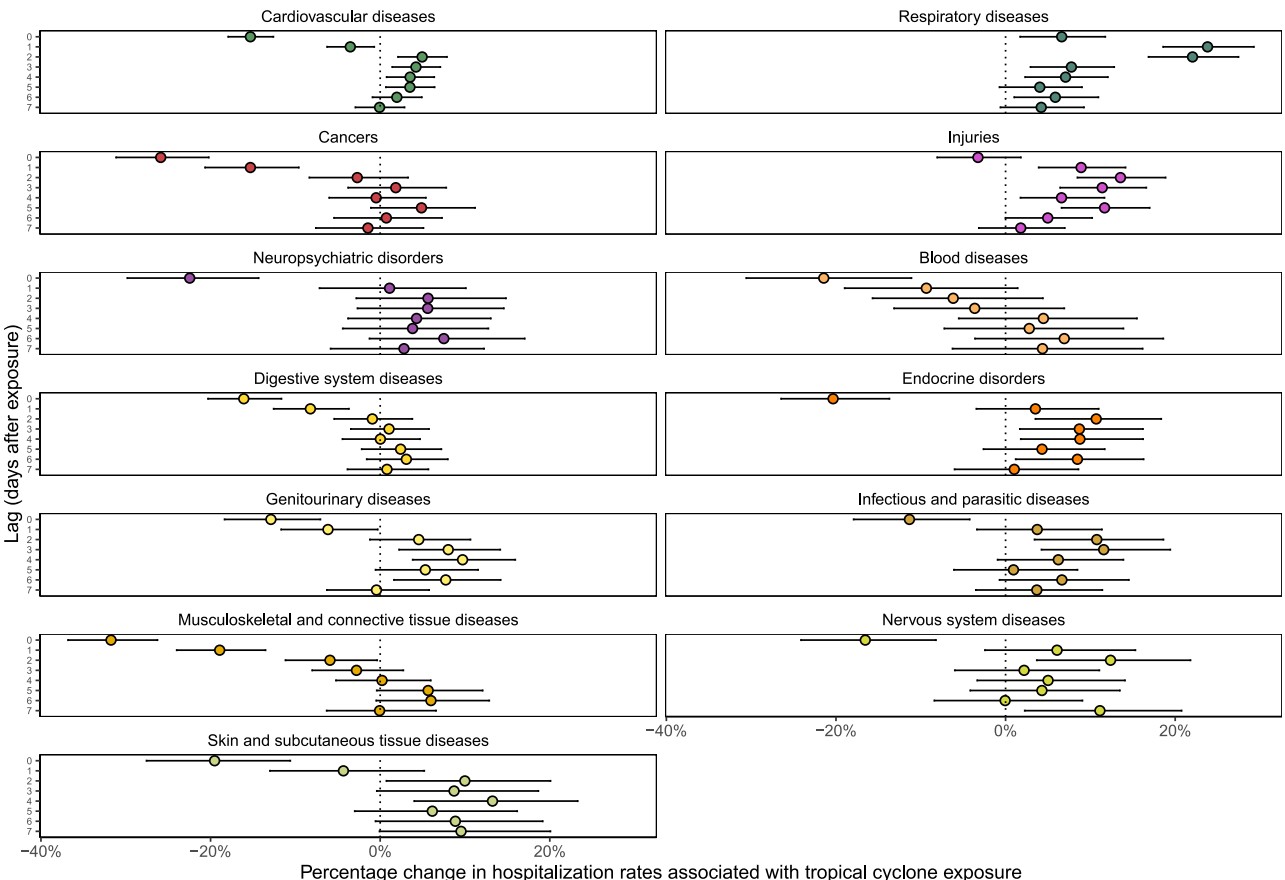

**Fig. 3 Percentage change in hospitalization rates with tropical cyclone exposure by cause of hospitalization and lag time.** Lag time is measured in days after tropical cyclone exposure. Dots show the point estimates and error bars represent Bonferroni-corrected 95% confidence intervals.

unmanageable without ventilators, nebulizers, or oxygen concentrators[41]. Because of the immediate risk to life, even the danger of leaving home in a storm or hurricane may not deter those seeking medical care for respiratory diseases.

**Injuries**. Injury hospitalizations are impacted by tropical cyclone exposure both directly and indirectly. During and immediately after tropical cyclones exposure, common injuries originate from transport accidents, structural collapse of buildings, wind-borne debris, falling trees, and downed power lines[32]. Days after exposure, other injuries such as puncture wounds, lacerations, falls from roof structures, chainsaw mishaps, and burns take more prominence in hospitalizations[32]. Though tropical cyclone wind exposure may bring about injuries in the Medicare population, those injured may not choose to seek treatment on a dangerous day to leave a property, which would explain the decrease in hospitalizations on the day of tropical cyclone exposure. This may be because the injury may not be life-threatening and the risk of getting caught in a tropical cyclone may be viewed as greater than not seeking treatment immediately[42]. There is also possibility of the indirect injuries occurring in the days following a tropical cyclone, e.g., during a clean-up process[6,43]. Houses and properties caught in a tropical cyclone may be severely damaged or nearly destroyed[2]; the subsequent clean up may present risks for hurting oneself accidentally, e.g., from electrocution[6].

**Cardiovascular diseases**. In our analyses, we did not observe overall changes in average cardiovascular disease hospitalization rates after tropical cyclone exposure. Specifically, we observed a decrease in hospitalization rates on the day of and day after

tropical cyclone, followed by a small increase over 2–6 days afterward. This contrasts previous studies reporting overall average increases, though those studies focused on case studies of single hurricane events[28,31]. For average changes in cause-specific hospitalizations within the broader cardiovascular disease cause group, we observed both positive and negative associations; acute myocardial infarctions (heart attacks) increased after tropical cyclone wind exposures, but non-acute cardiovascular hospitalizations, such as heart valve disorders, decreased. In a study associating snowstorms in Boston with cardiovascular disease hospitalizations, a similar lag pattern was observed[44]. Tropical cyclone exposure could also indirectly lead to increases in acute cardiovascular disease hospitalizations, due to increased stress and physical challenges brought about during and following exposure[6]. Disruption of access to essential medicines from closure of local supply sources, such as pharmacies, may also contribute to negative cardiovascular health outcomes[45,46]. Although we did not observe short-term changes (i.e., in first week after tropical cyclone exposure) in cardiovascular disease hospitalizations, longer-term negative impacts of tropical cyclone exposure on cardiovascular diseases have been observed, several years after exposure itself[47].

**Neuropsychiatric disorders**. Hospitalizations from neuropsychiatric disorders showed no overall average association with tropical cyclone exposure, though we observed an initial decrease of hospitalizations on the day of exposure. We observed an increase in delirium and dementia hospitalizations in the week after tropical cyclone exposure. There is evidence from studies of earthquakes in Japan that disasters can aggravate dementia, both

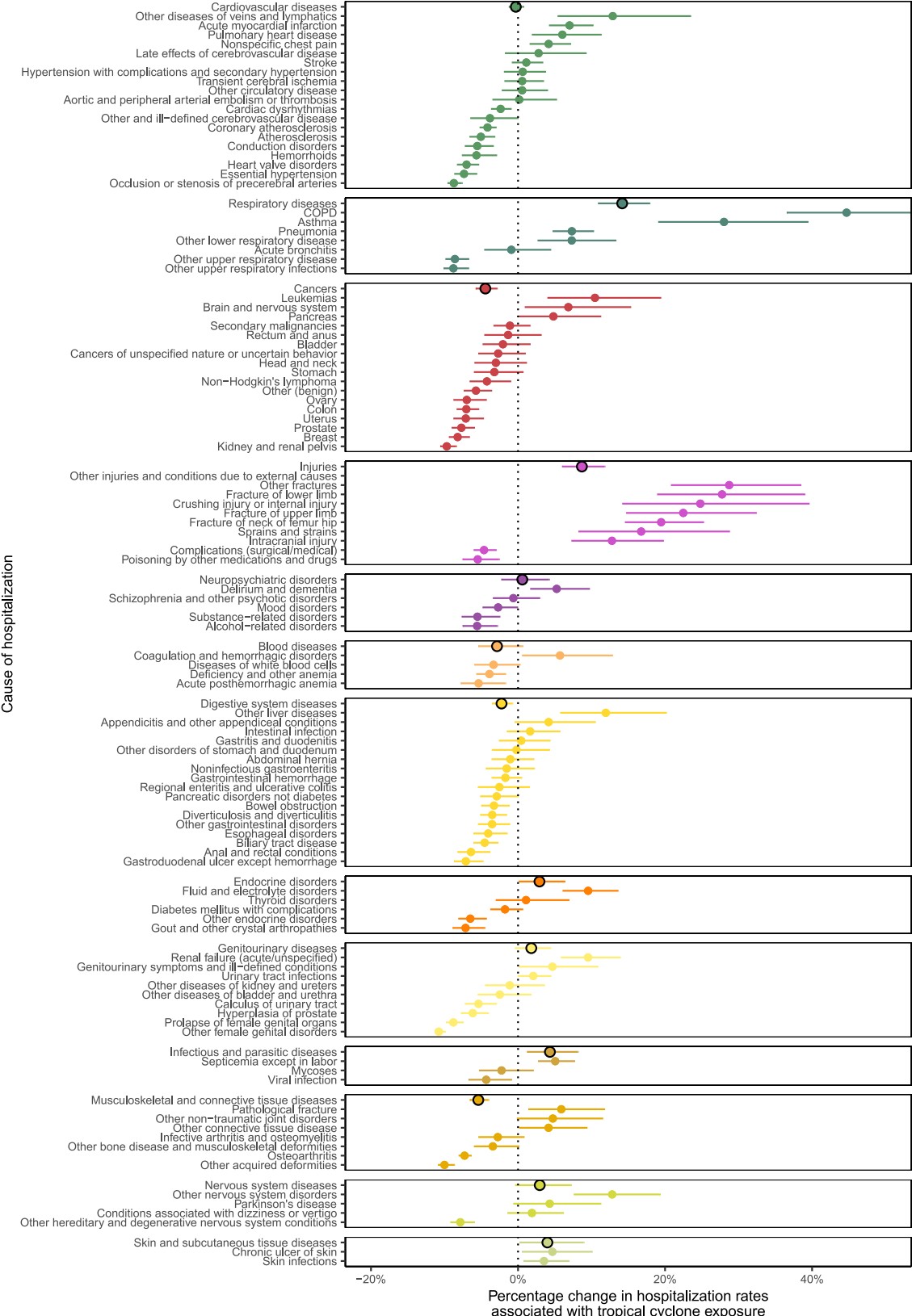

**Fig. 4 Average percentage change in hospitalization rates with tropical cyclone exposure by cause and sub-cause of hospitalization.** Average percentage change in hospitalization rates is across studied lag period (0–7 days after tropical cyclone exposure). Dots show the point estimates and error bars represent 95% confidence intervals.

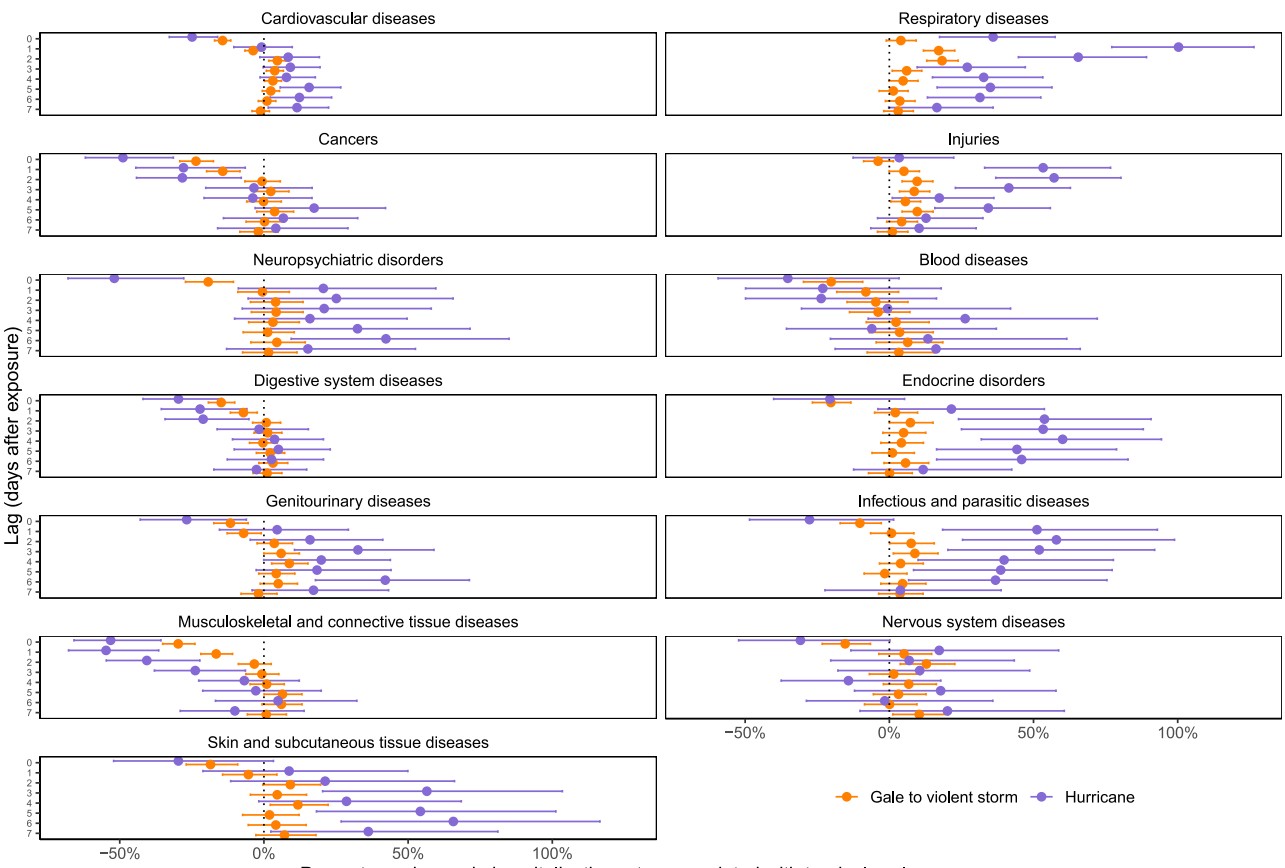

**Fig. 5 Percentage change in hospitalization rates with tropical cyclone exposure by cause of hospitalization, intensity of local wind exposure, and lag time.** Lag time is measured in days after tropical cyclone exposure. Dots show the point estimates and error bars represent Bonferroni-corrected 95% confidence intervals.

over short- and long-term periods; short-term increases in dementia hospitalizations may be due to moving vulnerable dementia patients out of care and nursing homes, which can cause stress from moving, while also providing worse care in evacuation sites[48]. Longer-term impacts after disasters may be due to the stress of a domestic property being damaged or destroyed. Tropical cyclone exposure can cause stress and anxiety following potential financial concerns, intimate partner violence, loss of property, loss of family and friends, and other sources of insecurity[28,40,49].

**Cancers**. We observed an overall reduction in cancer-related hospitalizations during the day of and week after tropical cyclone exposure. This association was largely driven by the reduction in cancer-related admissions on the day of and the day after exposure. Hospitalizations and treatment for cancers decrease in general in the aftermath of natural disasters, due to the damage to infrastructure, communication systems and medication, and medical record losses[50]. We observed different and distinct associations between tropical cyclone exposure and cause-specific hospitalizations within the broader cancer cause group. This may be due to more acute admissions needing immediate attention, such as leukemia or brain and nervous system cancer, while some prearranged admissions for patients with known cancers may be delayed[39]. For some cancers, lack of access to essential cancer medicine due to supply line disruption may compel a patient to travel to hospital[6].

**Endocrine disorders and genitourinary diseases**. An overall increase in endocrine disorder and genitourinary hospitalizations is plausible, as tropical cyclones can result in decreased availability of adequate food, water, and medicine, as well as electricity to store medicines properly[6]. Tropical cyclones can severely disrupt food and water supply lines, as well as close off medicine sources, either by locations closing temporarily or by discouraging those who need it from venturing into danger[6]. Electricity supplies which maintain medicines are often cut off during a storm or hurricane, at least temporarily[36]. Access to dialysis due to renal failure in the aftermath of a tropical cyclone would also rely on constant supply of electricity, which—when cut at home or unavailable at a local care provider—may result in additional hospitalizations for fluid and electrolyte disorders[51].

**Other diseases**. When a storm or hurricane passes through an area, stagnant and unclean water is often left behind[52,53], which can be optimal breeding grounds for many diseases, including infectious and parasitic, skin and subcutaneous, and blood and digestive system diseases[53,54], though absolute numbers are small compared with other hospitalization causes. Following the tornado in Joplin, Missouri, in 2011, unusual fungal skin infections were recorded[55]. Hospitalizations were caused by debris from the tornado infected with some fungus that is more common in uninhabited areas, but rarely makes it into humans' bodies. The increase in skin and subcutaneous disease may follow this pathway too, as of a result of cleaning up after the storm. Infectious diseases may also take time to be noticeable, as symptoms would only show up a few days after acquisition of an infection.

**Planning for tropical cyclone hospitalizations**. Other disasters, such as earthquakes and tsunamis, can also overwhelm hospital

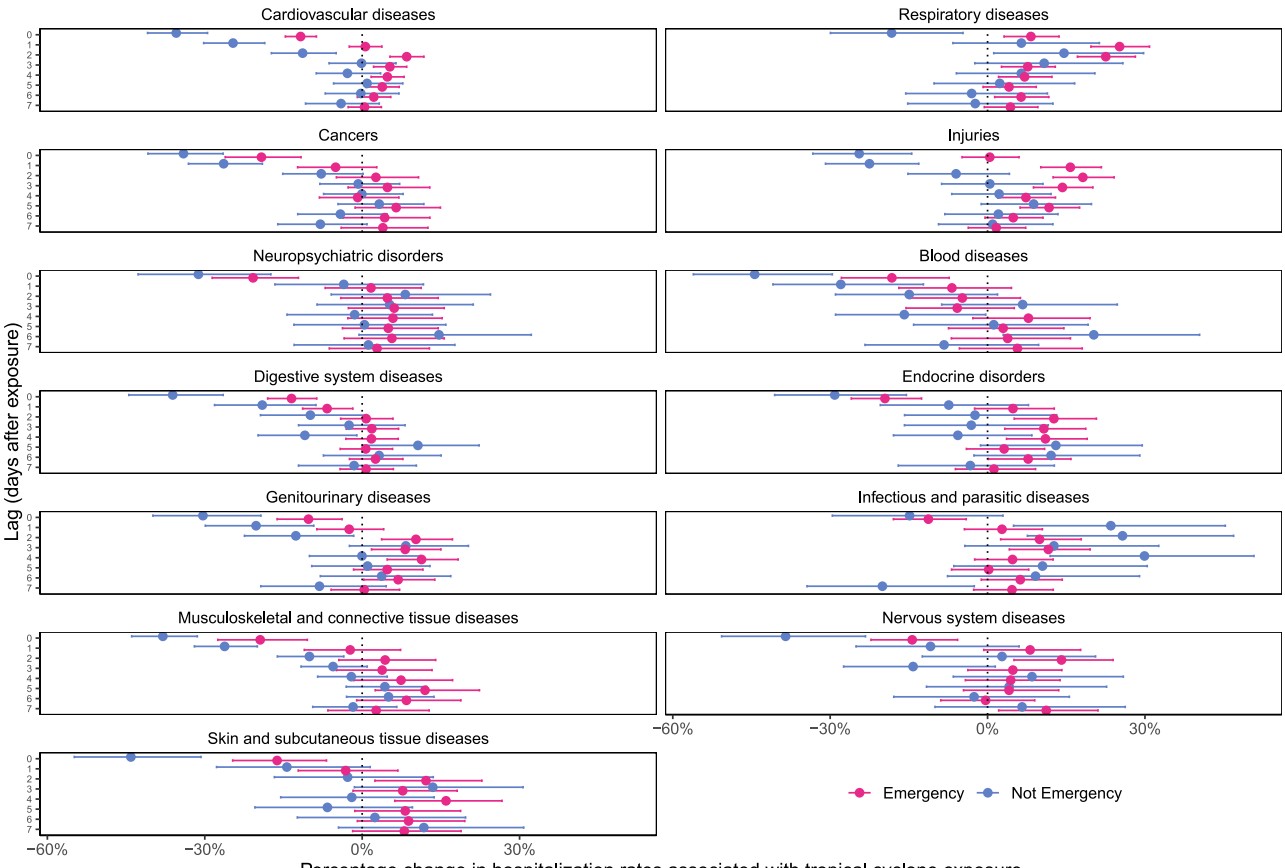

**Fig. 6 Percentage change in hospitalization rates with tropical cyclone exposure by cause of hospitalization, type of hospital admission, and lag time.** Lag time is measured in days after tropical cyclone exposure. Dots show the point estimates and error bars represent Bonferroni-corrected 95% confidence intervals.

capacity in a very short time[56,57]. Similar lessons of how to best assign hospital personnel from our work will be valuable to minimize patient suffering or, ultimately, death. While some cause-specific hospitalization rates may not change on average in the week following a tropical cyclone, the changing distribution of hospitalization rates during the post-cyclone week requires careful planning. Although many health care systems already have provisions in place, findings from our study may further inform planning.

**Strengths and limitations**. For our study, we leveraged hospitalization data from 70 million Medicare hospitalizations and linked those to a curated tropical cyclone exposure dataset in 30 states and districts over a 16-year period to comprehensively characterize the impact of tropical cyclones on cause-specific hospitalization rates in a vulnerable population. Nonetheless, our study also has some limitations. First, exposure misclassification is likely. Our results are based on patient county of residence; this may not necessarily be the location of the patient during a tropical cyclone. Furthermore, although we conducted analyses at the county level, tropical cyclone wind fields tend to be larger than the size of a county[58]. Any misclassification, nonetheless, is likely non-differential as it is not expected to be correlated with the outcomes assessed. Any resulting bias, therefore, would be toward the null[59]. Second, we also cannot exclude the possibility of some residual confounding. By design (matching), our study controls for all factors varying across counties, as well as month and season. We further adjusted for day of week, long-term trends, and temperature. Any variable inducing residual

confounding bias, thus, would have to covary with hospitalization rates and tropical cyclone exposure within county and be independent of the variables we have already incorporated in analyses. It is unlikely that our results are attributable to confounding bias.

**Future research**. We chose to focus on seniors, an already vulnerable population. Our results may not generalize to younger populations; further studies to investigate associations in different age groups are warranted. It will also be important to understand the differential impacts of tropical cyclones on health outcomes by geography, as well as socioeconomic and demographic factors. Further work is needed to specifically understand which hospitals would need to be prepared with the forecast of a tropical cyclone, along with which sources of health care are disrupted. Some negative health outcomes may also be acute and severe enough that those afflicted never get to hospital and die; in these cases, as with other disasters, such as earthquakes[60], rapid access to treatment is essential. There is some limited evidence to suggest that there are measurable long-term impacts on health in the years after a disaster[47,48]. There are plausible causal links between health outcomes and tropical cyclone exposure for many of the associations here[6,28–33], but more work needs to be done to identify and formalize these pathways. Characterizing longer-term health impacts of tropical cyclones is critical. We also focused on defining a tropical cyclone by wind speed, as it has direct relevance for identifying a tropical cyclone and therefore emergency planning[33,61]. Understanding in more detail whether including more information about specific tropical cyclone-related hazards, such as rainfall and flooding, in combination

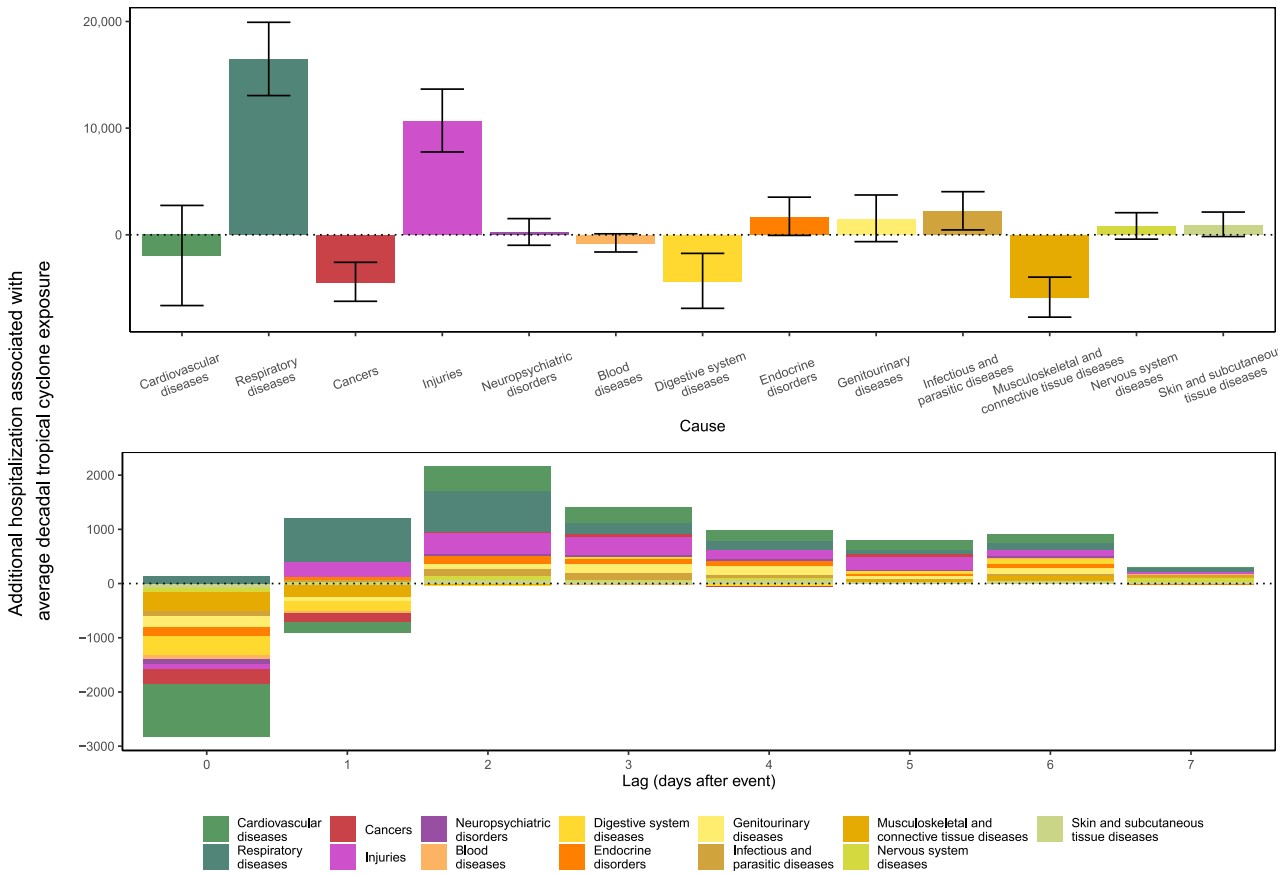

**Fig. 7 Change in hospitalizations for the US Medicare population, for an expected number of tropical cyclone exposures by county over a decade.** The top row shows the break down by cause covering the day of tropical cyclone exposure to 7 days afterward, with black bars representing Bonferroni-corrected 95% confidence interval. The bottom row shows the break down by lag days after the exposure.

with wind, modify the impact of tropical cyclones on health outcomes will be an important direction of future research. Our study included millions of hospitalizations over a decade across all counties impacted by tropical cyclones in the United States during this period. In more recent years, however, many catastrophic tropical cyclones have made landfall and future studies should include these data.

**Conclusion**. Our work provides valuable information on how cause-specific hospitalizations can be impacted by tropical cyclones, which can be used for preparedness planning, including hospital and physician preparedness. Adequate forecasting of tropical cyclones might help, for example, in the planning of setting up shelters to provide electricity and common medications and creating easy ways for vulnerable people with certain chronic conditions to find and use those resources outside of the hospital. While our study is the first step in identifying areas of improvement in hospital preparation, enabling and improving planning in this way would be a major innovation, which could save many lives of those who are hospitalized during tropical cyclones, and should be a public health priority.

## Methods

**Study population**. We obtained Medicare inpatient claims data from the Center for Medicare and Medicaid Service (CMS) and assembled data from Medicare beneficiaries, aged 65 years or older, enrolled in the fee-for-service program for at least 1 month from January 1, 1999, to December 31, 2014, and residing in the United States. For each enrollee, we extracted county and state of residence from the Medicare enrollee record file, and the date and principal ICD-9-CM code for each hospitalization from the Medicare Provider Analysis and Review (MEDPAR) file. We restricted analyses to counties that experienced at least one tropical cyclone

exposure during the study period. Hospitalizations were aggregated to the county level based on the patient's county of residence, and included both emergency and nonemergency hospitalizations.

This study was approved by the Institutional Review Board at the Harvard T.H. Chan School of Public Health.

**Outcome assessment**. We grouped the 15,072 possible ICD-9-CM codes by the CCS hierarchy algorithm[38], developed by the Agency for Healthcare Research and Quality, into 18 mutually-exclusive and clinically-meaningful CCS level 1 disease causes. We excluded five causes that occur rarely in older adults, such as pregnancy or fertility issues, and those that were ill-defined. This left 13 level 1 causes, accounting for 94.8% of recorded hospitalizations. These 13 level 1 causes were: cardiovascular diseases, respiratory diseases, cancers, injuries, neuropsychiatric disorders, blood diseases, digestive system diseases, endocrine disorders, genitourinary diseases, infectious and parasitic diseases, musculoskeletal and connective tissue diseases, nervous system diseases, and skin and subcutaneous tissue diseases. We additionally investigated associations with CCS level 3 causes; to avoid unstable model outputs, we restricted this secondary analysis to sub-causes with >50,000 total hospitalizations across all counties during the study period.

**Exposure assessment**. We obtained data on tropical cyclone wind exposure in the United States from the hurricaneexposure (version 0.1.1) and hurricaneexposuredata (version 0.1.0) packages in R, with full space and time coverage over our study period, and described in detail elsewhere[58,62–64]. In brief, an exhaustive assessment of tropical cyclones was generated from those recorded in the HUR-DAT2 dataset based on wind field modeling and validation against observations from weather stations[65]. First, all tropical cyclones were measured for how close they came to at least one US county. Cyclones that came within 250 km were retained for further wind modeling[58]. For these, the wind field at each county's population mean center was modeled every 15 min, while the storm was tracked, providing an estimate of peak local winds that the storm brought to that county[66]. This modeling used a double exponential wind model to estimate 1-min surface wind at each county center, based on the storm's forward speed, direction, and maximum wind speed[66,67]. We used daily estimates of maximum wind sustained speed by county to generate classifications of these exposures. This tropical cyclone wind exposure dataset covered the 898 counties in 30 state or district units

(29 states and Washington DC) with at least one tropical cyclone wind exposure during 1999–2014.

From continuous estimates of local storm-associated winds, we classified a county as exposed to tropical cyclone winds based using a cut point from the Beaufort wind scale[68]. The Beaufort scale is an empirical measure that relates locally measured wind speed to observed conditions on sea or land from 0 (calm) to 12 (Hurricane force). In contrast, the Saffir–Simpson wind scale provides a classification for storm-wide, rather than local, wind intensity[69]. Our primary analysis focused on tropical cyclone winds, i.e., ≥34 knots, which include both hurricanes and tropical storms. We defined tropical storm exposure when the peak sustained wind that day in the population center of the county associated with the tropical cyclone reached or exceeded 34 knots (63 km/h, 39 mph; gale-force wind on the Beaufort scale) up to 64 knots (119 km/h, 74 mph; violent storm-force wind on the Beaufort scale), when the tropical cyclone was at the point of closest approach to that county. We defined hurricane exposure when the peak sustained wind that day in the population center of the county associated with the tropical cyclone reached or exceeded 64 knots (119 km/h, 74 mph; hurricane-force wind on the Beaufort scale), when the tropical cyclone was at the point of closest approach to that county. In secondary analysis, we used a three-category exposure variable (unexposed, gale to violent storm wind exposure, and hurricane wind exposure).

**Covariate data**. We obtained data on temperature from the Parameter-elevation Regressions on Independent Slopes Model (PRISM), which gathers climate observations from a wide range of monitoring networks and applies sophisticated quality control measures to generate a nationwide temperature dataset, with full space and time coverage over our study period[70]. We used gridded daily estimates at a resolution of $4 \, km^2$ to generate area-weighted daily temperatures by county.

**Statistical analysis**. We analyzed the association between daily hospitalization rates and tropical cyclone exposure by applying a conditional quasi-Poisson model[71]. The quasi-Poisson formulation accounts for potentially overdispersed outcomes. This approach examines contrasts within matched strata, similar to a case-crossover study design, thus eliminating any confounding bias that could arise by factors varying across strata in a computationally efficient way[71]. Specifically, we matched on county and Julian day of the year, controlling for non-time-varying factors varying across counties in our analyses, as well as seasonality. We flexibly adjusted for longer-term time trends in factors that varied over our study period, and could covary with both tropical cyclone exposure and hospitalization rates. We also adjusted for the mean temperature at the day of the tropical cyclone exposure and day of week. Finally, we included in this model unconstrained distributed lag terms for the exposure[72,73], to quantify the association between tropical cyclone exposure and hospitalization rates up to 7 days after exposure. Specifically, we fit the following model:

$$\log\left(E\left[Y_{ct}\right]\right) = \alpha_0 + \alpha_{ct} + \sum_{l=0}^{7} \beta_l \text{Exposure}_{lct} + \beta_1 \text{Temperature}_{ct}$$
$$+ \sum_{d=1}^{6} \beta_d \text{DOW}_t + ns\left(\text{year}_t\right) + \log\left(\text{Population}_{ct}\right),$$

(1)

where $Y_{ct}$ denotes the number of CCS level 1-defined cause-specific hospitalizations in county $c$ and day $t$; $a_{ct}$ the stratum-specific intercepts (not estimated in the conditional Poisson model); $\beta_l$ lag-specific coefficients (log rate ratios) for tropical cyclone exposure, with $l \in [0, 7]$ lags between the day of tropical cyclone exposure and day of hospitalization and $\Sigma \beta_l$ the cumulative effect of tropical cyclone exposure on cause-specific hospitalizations over eight days (lags 0–7); $DOW_t$ the day of week; $ns(\text{year}_t)$ a natural spline with two degrees of freedom to flexibly model time trends (seasonal trends are captured through matching); and $\log\left(\text{Population}_{ct}\right)$ the offset with population the number of Medicare enrollees in each county and year. We applied the Bonferroni–Holm method to adjust CIs for multiple comparisons[74]. The 95% CIs were corrected by using $\alpha = 0.05/D$, where $D = 13$, the number of causes in our main analysis[75].

We assessed whether estimated effects varied for emergency vs. nonemergency CCS level 1 hospitalization causes by fitting stratified analyses by admission type, using the same model as described above. In secondary analyses, we evaluated associations between tropical cyclone exposure and CCS level 3 cause-specific hospitalizations. We present the CCS level 3 associations as the average change in cause-specific hospitalization rate over lags 0–7, i.e., $\Sigma \beta_l / 8$. We used a three-category exposure term to estimate independent effects of tropical cyclone wind exposures separated into gale-force to violent storm-force and hurricane-force intensities for CCS level 1 causes.

Finally, we used the cumulative rate ratio estimates and average cause-specific hospitalization rates during May to October for each cause across 1999–2014, to calculate the expected number of excess (or fewer) hospitalizations during the week following the expected number of tropical cyclone exposures by county over a decade. Specifically, we multiplied the observed average weekly hospitalization rates for each county in May to October across 1999–2014 by the corresponding population of Medicare enrollees, $\left(\exp\left(\Sigma \beta_l\right)\right)^n - 1$ (ref. [73]), where $n$ is the average number of tropical cyclone exposures per year in each country times ten (number of years in a decade).

We present all results as percentage changes in hospitalization rates, unless otherwise noted. We conducted all statistical analyses using the R Statistical Software, version 3.6.3 (Foundation for Statistical Computing, Vienna, Austria)[76]. To specify the models, we used the gnm function from the gnm package, version 1.1-1 (refs. [71,77]). We also used the ns function from the splines package, version 3.6.3 (ref. [78]).

**Sensitivity analyses**. We assessed the sensitivity of our results to temperature adjustment. We fit models (1) including the described temperature term, as well as temperature terms of up to 7 days after exposure and (2) with no temperature term. Our results were robust to these sensitivity analyses.

## Data availability

Tropical cyclone exposure data are publicly available via the R packages hurricaneexposure (version 0.1.1) and hurricaneexposuredata (version 0.1.0) [https://github.com/geanders/hurricaneexposuredata/blob/master/data/storm_winds.rda], based on tropical cyclones recorded in the HURDAT2 dataset [https://www.aoml.noaa.gov/hrd/hurdat/Data_Storm.html]. Medicare enrollees dynamic cohort data are publicly available, upon purchase and after an application process, from the Centers for Medicare & Medicaid Services (CMS) [https://www.cms.gov/Research-Statistics-Data-and-Systems/CMS-Information-Technology/AccesstoDataApplication/index].

## Code availability

All code for analysis and visualization presented in this manuscript is available at www.github.com/rmp15/tropical_cyclones_hospitalizations_nat_comms.

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

## Acknowledgements

R.M.P. was supported by the Earth Institute postdoctoral research fellowship at Columbia University. F.D. was funded by the Climate Change Solutions Fund. Funding was also provided by the National Institute of Environmental Health Sciences (NIEHS) grants R01 ES030616, R01 ES028805, R01 ES028033, R01 MD012769, R01 AG066793, R01 ES029950, R21 ES028472, P30 ES009089, and P42 ES010349. The computations in this paper were run on the Research Computing Environment (RCE) supported by the Institute for Quantitative Social Science in the Faculty of Arts and Sciences at Harvard University. We thank Ben Sabath and Danielle Braun for assistance with computational challenges, Jane W. Baldwin for discussions on tropical cyclones, and James E. Bennett for discussions on the statistical model.

## Author contributions

All authors contributed to study concept and interpretation of results. R.M.P. collated and organized hospitalization files. R.M.P. collated and organized the storm and hurricane data from the dataset provided by G.B.A. R.M.P., G.B.A., and M.-A.K. developed the statistical model, which was implemented by R.M.P. R.M.P. performed the analysis, with input from M.-A.K., G.B.A., R.C.N., and F.D. A.N.-A. assisted with interpretation of results. R.M.P. and M.-A.K. wrote the first draft of the paper; all authors contributed to revising and finalizing the paper.

## Competing interests

The authors declare no competing interests.
