## [Peer Review File · Nature Communications]

Reviewers' Comments:

Reviewer #1:

Remarks to the Author:

General comments: Several potential limitations for consideration.

1. There are a number of studies in this field that are not cited, including prior studies using claims data and Medicare data to measure impacts on health care cost and utilization after disasters.
2. Misclassification of exposure is always a concern when the exposure is measured at the county level and the outcome is individual. How do we know that those who go to the hospital were the same who were actually exposed.
3. Most studies in this field only assess the impacts of a single disaster in each location. By including only counties with a single disaster event there can be a clear follow-up period and reduced confounding by multiple types of disaster events, however, this does reduce power.
4. Difference in differences has also been used fairly widely in these studies due to their susceptibility to confounding by unmeasured differences between the exposed and unexposed populations. Was this considered?
5. Were any Census or other covariates (e.g., percent renter-occupied units, median household income, percent of persons who speak English less than well and percent of adults with more than high school education) used for adjustment? These variables have been used previously in developing social vulnerability indices and controls in county-level studies.

Specific comments on manuscript:

Abstract: Although there is more to be done, I would not say that there are limited studies on the health outcomes from disasters. There are academic studies as well as a larger grey literature related to data from regular public health surveillance and enhanced surveillance (e.g., in shelters) after disasters. A major challenge of all disaster research is exposure misclassification. With exposure assigned at the county level, and health outcomes at the individual level, it is not possible to know if the hospitalized individual was actually exposed. Yes, targeted preparedness strategies but also Medicare and other systems should improve preparedness.

Introduction: There are more factors than climate change – subsidence, increases in the proportion of impervious surfaces – the damage in terms of dollars is also only one piece of the severity measure as we have increased the investment at risk as more populations (and the concomitant infrastructure to manage them) have moved to locations at higher risk of a natural disaster.

Page 3: CVD, dialysis should be mentioned

I feel that the last paragraph of the introduction could be deleted.

Results: Again, there is a potential problem with exposure misclassification.

Page 4: Most studies only allow for a single disaster exposure – what is different about counties (and their populations) who have multiple exposures over the study period?

Page 5: Could this peak be capturing substituted hospitalization for care missed due to the disaster, such as home health, in patient rehab, MD offices closed so only place for care? The decline in cancer hospitalizations would seem to be related to missed appointments – some cancelled by patients and others by providers. There are several papers that address this (Radcliff et al. using VA data)

Page 5: What does a 100% increase mean? What are the raw numbers? Is there a way to report rates as part of the numbers in Lines 127-134?

Discussion: People may be deterred from seeking care if evacuation orders are issued, if EMS services are suspended for non-evacuees. There is an issue with just identifying health outcomes with no real plausible relationship with intensity or duration of disaster exposure (e.g., leukemia).

Page 8: There are most definitely cases where disaster exposure prevents normal care management. Please see POST KATRINA

Page 9: There are data on most frequent injuries and time periods in which they are most likely.
Page 9: The multiple mentions of supply lines for essential medications seems a stretch. In 1 or 2 days? Tropical storms have highly improved forecasting and warning time in days, not hours or minutes. Supplies can be stockpiled and supply lines and logistics systems fortified in advance.
Page 10: Stress is the result of far more than property damage or loss of life. Financial concerns, intimate partner violence, insecurity of all types is exacerbated by a disaster's impacts.
Page 11: Line 245: Missed dialysis should be mentioned. Several papers:
It should be mentioned that the number of infectious diseases are relatively small. Again, shelter surveillance for respiratory and GI conditions to address possible outbreaks.
Line 257: No need to bring COVID to this paper. Start this paragraph with Disasters, such as...the other statements are an overreach.
Line 261: Health care systems have many provisions in place such as shelter in place to ensure personnel are in place.
Page 12, Line 283: This study doesn't address capacity. In natural disasters, hospital emergency capacity is also another literature. Should not be mentioned here. Again, the number of deaths directly caused by disasters in the U.S. is relatively small.
Page 13: We have highly improved and relatively accurate forecasting. Telemedicine and remote prescription is again an over reach. Not at all clear how this is relevant to hospitalization during tropical cyclones.

Reviewer #2:

Remarks to the Author:

Parks and colleagues leveraged a large Medicare dataset in the US with 70 million hospitalizations over 16 years to examine how tropical cyclone wind exposures affect hospitalizations from 13 disease categories and common sub-categories. They found that tropical cyclone exposure was associated with increases in hospitalizations from respiratory diseases, infectious and parasitic diseases, and injuries, but decreased cancer-related hospitalizations and no consistent changes in cardiovascular disease hospitalizations. This study is by far the most comprehensive study investigating the effects of tropical cyclone exposures on cause-specific hospitalizations. The study is well-conducted in terms of both exposure assessment and statistical methods. I enjoyed reading this manuscript. The discussions on both biological plausibility and limitations are remarkably clear and well acknowledged. Thus, I only have a few minor comments for the authors to consider in improving this paper.

Specific comments

1. Is it possible to compare the difference in hurricane-related risks between emergency room visits and other hospitalizations? Given the decreases in hospitalizations from cancer and some specific cardiovascular diseases, and the authors' argument that chronic and "non-emergency procedures being delayed or rescheduled," it would be interesting to see if the hurricane exposures would increase cancer or cardiovascular hospitalizations transferred from emergency room visits.
2. Please briefly justify the reason for choosing wind speed as the exposure metric for tropical cyclones, but not other metrics such as rainfall or flooding. Are the wind field modeling data validated against observations from weather stations?
3. In the primary statistical model, why not control for potential confounding from relative humidity?
4. Page 6, Lines 136-137. Please briefly describe the sensitivity analyses and associated findings here.
5. Lines 294-295. This statement is not based on the results of this study. During storms that are

often leading to power outage or Internet connection lost, telemedicine might not be a good choice.

Reviewer #3:

Remarks to the Author:

Tropical cyclone exposure is associated with increased hospitalization rates

Thank you very much for the opportunity to review the manuscript entitled, Tropical cyclone exposure is associated with increased hospitalization rates, submitted for consideration for publication in Nature Communications.

This paper makes an important contribution to the field by presenting a cogent and consistent analysis of the patterning of 70 million Medicare hospitalizations over a 16-year period of analysis. An innovative approach has been devised for examining tropical cyclone exposure based on wind field modeling in order to comprehensively map counties experiencing at least one day of gale force winds associated with tropical systems during the 16-year analysis window that could then be matched to Medicare hospitalizations over the subsequent 7-day period.

This yielded an array of Gulf of Mexico and eastern seaboard counties, along coastlines but also extending far inland because these tropical systems retained their strong wind hazards, and sometimes, their cyclonic signatures, for prolonged periods as they moved over large geographic expanses. This was a brilliant decision that provides a comprehensive look at tropical cyclone impact on population health, at least in the one dimension analyzed, patterns of Medicare hospitalization. The massive size of the data sampled permitted a nuanced presentation of hospitalization patterns that rationally fit the realities of storm impacts over multiple hurricane seasons.

The series of figures showcases the results in a vivid manner that allows the reader/viewer to easily understand the patterns of hospitalization by disease category and days post-impact. This is an elegant portrayal of the data.

Recommendations for consideration:

Please consider specifying in the title that the focus is on Medicare hospitalizations. This represents a very important subset of inpatient admissions. This is an older subpopulation with much higher utilization rates and prevalent morbidities and co-morbidities. Tropical cyclones alter their utilization patterns as clearly depicted. Nevertheless, patterns of injury and health services utilization for storm-related conditions are likely to be quite different for younger residents in the same storm-affected areas. The specificity of this paper, focusing on the Medicare population, is clarified right away in the abstract but should also be conveyed in the title.

There is a cursory mention of anthropogenic climate change in the introduction along with a cluster of citations. Several important papers, including several by Kossin might wisely be added. Also, the Knutson paper (#15) in BAMS is actually Part II of a 2-part series. Consider citing Part I also, as it is equally or more relevant.

It might be wise to add or expand on two points in the discussion. Please consider:

First, please explain why were analyses presented only through 2014? Some of the most remarkable and active storm seasons have occurred from 2015 through 2020. These were the years that included such memorable storms as Matthew, Harvey, Irma, Maria, Florence, Michael, and Dorian.

Second, especially with progressively increasing influences of climate change on tropical cyclone behavior, the water hazards (storm surge, coastal wave action, extreme rainfall totals and precipitation rates, widespread inland freshwater flooding) are equally prominent hazards that both increase demand for hospitalization but also impede access. Take the example of Hurricane Harvey in 2017 (outside your window of analysis but well known). Winds quickly died down to below tropical storm force but with the center of circulation remaining near-stationary over the western Gulf, the storm unleashed 33 trillion gallons of rain on Texas and a portion of Louisiana over five days (with rainfall totals over 60 inches in some areas). Same for Florence over the Carolinas in 2018. This might be worth a mention. If data sets are available, this might make for an intriguing future analysis.

Authors are wise to indicate that future analyses should be extended to persons who are younger than the Medicare population.

Because of the focus on inpatient hospitalization and the decision to limit the timeframe to seven days post-impact, neuro-psychiatric hospitalizations show minimal changes except for a blip of delirium and dementia cases. Nonetheless, outpatient and inpatient consultation for cyclone-associated PTSD and mood disorders are likely to become an important feature of health care utilization starting 5-6 weeks following impact. Optional, but might be worth a comment when you discuss this cluster of diagnoses.

Finally, the tone of your commentary about what you are observing for each disease category "sounds" a bit speculative and uncertain as I read what is written. For example, you speculate about the dramatic surge in respiratory admissions for disorders like COPD (CLRD) and mention a bit hesitantly that perhaps this could be due to power outages. This is almost certainly the case. As one example, the frequently-hit state of Florida has a designated special needs shelter ("SpNS") in every county and a list of persons registered for pick up and transport to this retrofitted facility that is equipped with a massive, elevated (above flood waters) auxiliary generator that maintains power for days when local electrical service in the area is completely offline. Inside, these electrically-dependent (typically oxygen-dependent) local residents are able to plug in their concentrators and maintain the life-giving flow of oxygen even as the storm is raging outside (this reviewer has been onboard a SpNS during multiple hurricanes and the system works well). I add this lengthy comment to say, the one voice that seemed to be missing was that of someone who has considerable direct, onsite storm experience to provide a more confident tone to your interpretations.

This is an excellent paper and an important addition. Thank you for dedicating your time to preparing and clearly presenting your analysis.

James M. Shultz

We thank the Editors and Reviewers for their thoughtful and constructive suggestions. We have revised the manuscript and added sensitivity analyses in response to the Reviewers' comments, as detailed below.

All page/line/reference numbers refer to the tracked manuscript.

Reviewer 1

General comments: Several potential limitations for consideration.

We thank the Reviewer for the thoughtful and constructive suggestions. We have responded point-by-point to the Reviewer's questions and comments below.

1. There are a number of studies in this field that are not cited, including prior studies using claims data and Medicare data to measure impacts on health care cost and utilization after disasters.

We have added some previous studies in the Introduction in the revised manuscript to better represent existing literature (P. 3, Lines 53-58):

Some studies have reviewed general evidence of health impacts of storms and hurricanes, primarily using case studies, for cardiovascular diseases, respiratory diseases dialysis-, and injury-related hospitalizations, showing harmful impacts overall.^{6,28-33} Other studies have previously used claims and Medicare data to measure impacts of disasters, including how mortality and morbidity was affected in the Medicare population after Hurricane Katrina³⁴ and changes in Medicare cost and utilization.³⁵

2. Misclassification of exposure is always a concern when the exposure is measured at the county level and the outcome is individual. How do we know that those who go to the hospital were the same who were actually exposed?

The Reviewer is certainly right to point out that exposure misclassification is commonly raised as a worthy consideration for large-scale epidemiologic studies such as ours. Our group, in fact, is committed to quantifying and correcting for exposure measurement error in health studies, primarily in air pollution applications.(Hart et al., 2015; Kioumourtzoglou et al., 2014; X. Wu et al., 2019; Zeger et al., 2000) Non-differential misclassification of binary exposure would bias results towards the null.(Carroll, Ruppert, Stefanski, & Crainiceanu, 2006) There is no reason to believe that exposure misclassification in this study would be correlated with the number of events per county, as exposures were independently estimated. Therefore, any misclassification of exposure would indicate that our estimates in Figures 3-5 would be an underestimate, if anything. We would also like to clarify that we did not assess the outcome at the individual level. The dependent variable in all analyses was the number of cause-specific daily events per county. Our aim, therefore, was to adequately capture county-wide average population exposure.

We are confident that our exposure classification for counties over space and time are accurate for several reasons, as reported in a research paper in press by our group.(Anderson et al., 2020) First, wind fields of tropical cyclones tend to be much larger than the size of a county. Therefore, the majority of the counties in our study will have been clearly either completely within the wind field or completely outside. Second, wind speeds evolve slowly moving out from the storm center, except if very near the coast or very near the center of the storm. In those

cases, it will be unusual for the winds to be near that 34-knot threshold. Instead, they would usually be well above the threshold and, therefore, the county would be classified as exposed anyway.

Further, we agree with the Reviewer that we cannot guarantee in our study that those who go to the hospital were the same who were exposed. This is a known source of potential misclassification in all observational environmental epidemiologic studies that do not have information on the location of all cohort participants at each time point. Given the massive amount of data we leveraged for this analysis, it would not have been feasible to collect information for all Medicare enrollees' exact locations each day over the 16 study years. However, this would also bias our results to the null, resulting in an underestimate of the association.

We thank the Reviewer for raising these valid issues, and we have elaborated on these points to the limitations of the revised manuscript (P. 15, Lines 349-354):

First, ~~we~~ exposure misclassification is likely. Our results are based on patient county of residence; this may not necessarily be the location of the patient during a tropical cyclone. Furthermore, although we conducted analyses at the county level, tropical cyclone wind fields tend to be larger than the size of a county.⁵⁸ Any misclassification, nonetheless, is likely non-differential as it is not expected to be correlated with the outcomes assessed. Any resulting bias, therefore, would be towards the null.⁵⁹

3. Most studies in this field only assess the impacts of a single disaster in each location. By including only counties with a single disaster event there can be a clear follow-up period and reduced confounding by multiple types of disaster events, however, this does reduce power.

We agree that most studies in the field have hitherto assessed the impacts of a single disaster in each location; this in fact formed part of the stimulus for carrying out the research in our study (PP. 2-3, Lines 39-42). Our goal was to investigate the average impact of exposure to tropical cyclones on a comprehensive list of hospitalization causes, including less intense tropical cyclones, the impacts of which have not been previously thoroughly evaluated, as they are always universally recognized as disaster events. Furthermore, our distributed lag model can accommodate multiple tropical cyclone events and has previously been used to model the impacts of heat waves (Bobb, Obermeyer, Wang, & Dominici, 2014) and snowstorms (Bobb et al., 2017) on adverse health outcomes. Because the model estimates a different effect for each lag day, it can accommodate a day that, for example, is two days after one exposure event and also four days after another (P. 28, Lines 688-690).

4. Difference in differences has also been used fairly widely in these studies due to their susceptibility to confounding by unmeasured differences between the exposed and unexposed populations. Was this considered?

We agree with the Reviewer that confounding by unmeasured differences between exposed and unexposed populations in an observational study is always a concern. We also agree that the Difference in Difference (DID) approach is one way of handling such issues. However, DID approaches (1) require a comparison between an exposed population and a different, but exchangeable, unexposed population and, importantly, (2) also assume no unmeasured confounding. (Wing, Simon, & Bello-Gomez, 2018) With our approach, using a conditional

quasi-Poisson model matching on county,(Armstrong, Gasparrini, & Tobias, 2014) we only contrast an exposed county with itself at a comparable time, in this case by also matching on Julian day of year, thus fully adjusting for month and season. By this matching structure, we are, therefore, fully controlling for all confounding by factors that vary across counties and seasonal trends. We additionally adjusted for other potential temporal confounders, including day of week and long-term trends. For this reason, our design, for this analysis, is more appropriate and robust than a DID approach.

5. Were any Census or other covariates (e.g., percent renter-occupied units, median household income, percent of persons who speak English less than well and percent of adults with more than high school education) used for adjustment? These variables have been used previously in developing social vulnerability indices and controls in county-level studies.

For this analysis, we controlled for any factors that would vary across counties by matching on county. Specifically, we used a conditional quasi-Poisson model,(Armstrong et al., 2014) where we only contrast an exposed county with itself at another comparable time, effectively accounting for any exogenous or endogenous differences across counties. Therefore, Census or other covariates that vary across counties cannot induce confounding in this study design. Furthermore, we adjusted for long-term trends in the statistical model, also effectively controlling for any time trends in these variables within counties.

Specific comments on manuscript:

Abstract: Although there is more to be done, I would not say that there are limited studies on the health outcomes from disasters. There are academic studies as well as a larger grey literature related to data from regular public health surveillance and enhanced surveillance (e.g., in shelters) after disasters. A major challenge of all disaster research is exposure misclassification. With exposure assigned at the county level, and health outcomes at the individual level, it is not possible to know if the hospitalized individual was actually exposed. Yes, targeted preparedness strategies but also Medicare and other systems should improve preparedness.

We agree with the Reviewer that there are not limited numbers of previous studies, but in reference to hospitalizations and tropical cyclones, our study is a major step forward due to the exhaustive nature of our datasets. We have modified the language in the abstract (P. 2, Lines 16-19):

Hurricanes and other tropical cyclones have devastating effects on society. ~~Previous Limited~~ studies have quantified their impact on ~~several some~~ non-fatal health outcomes. Here, we used data on 70 million Medicare hospitalizations and tropical cyclone exposures over 16 years (1999–2014).

We understand the Reviewer's concern about exposure misclassification. Please see our detailed response regarding exposure misclassification in our study and expected direction of bias in the Reviewer's General Comment #2 above.

Introduction: There are more factors than climate change – subsidence, increases in the proportion of impervious surfaces – the damage in terms of dollars is also only one piece of the severity measure as we have increased the investment at risk as more populations

(and the concomitant infrastructure to manage them) have moved to locations at higher risk of a natural disaster.

We thank the Reviewer for this suggestion. We have updated the revised manuscript accordingly (P. 2, Lines 35-37):

The intensity of tropical cyclones is predicted to change, ~~along with their impacts on health,~~ due to anthropogenic climate change.¹⁰⁻¹⁵¹⁰⁻¹⁹ Land subsidence²⁰ and increases in the proportion of impervious surfaces²¹ may further exacerbate cyclone impacts.

Page 3: CVD, dialysis should be mentioned.

We thank the Reviewer for this suggestion. We have updated the revised manuscript accordingly (P. 3, Lines 53-55):

Some studies have reviewed general evidence of health impacts of storms and hurricanes, primarily using case studies, for cardiovascular diseases, respiratory diseases dialysis-, and injury-related hospitalizations, showing harmful impacts overall.^{6,28-33}

I feel that the last paragraph of the introduction could be deleted.

We thank the Reviewer for the suggestion. With the last paragraph in the Introduction, we justify the need for our study and what knowledge gaps we think it helps address. We also explicitly state our aim and put our research in the context of climate and health research. However, in response to the Reviewer's previous comments and in agreement with the revised Introduction section, we have removed the word 'substantial' (P. 4, Lines 65-67):

Despite these prior findings and biological plausibility, there is an overall ~~substantial~~ knowledge gap in consistently and comprehensively quantifying how tropical cyclone exposure drives hospitalizations across time and space.

Results: Again, there is a potential problem with exposure misclassification.

We would refer the Reviewer to our detailed response to the Reviewer's General Comment #2 on exposure misclassification.

Page 4: Most studies only allow for a single disaster exposure – what is different about counties (and their populations) who have multiple exposures over the study period?

Counties along the coast were generally exposed more than those inland (manuscript Figure 1), with the largest overall number of exposures in North Carolina (P. 4, Lines 80-84). We agree with the Reviewer that it is important to investigate the differential impacts of tropical cyclones on health outcomes, for example by geography and by socio-economic and demographic factors. Ours is the first exhaustive study of the association between tropical cyclones and numerous cause-specific hospitalizations; our primary analyses investigate associations with 13 classes of hospitalization causes and our secondary analyses with more than 100 causes. Further evaluating potential effect modification by socio-demographic factors, thus, is beyond the scope of this analysis. However, future studies should certainly explore these. We have provided more comment in the Discussion section of the revised manuscript (P. 16, Lines 362-367):

Our results may not generalize to younger populations; further studies to investigate associations in different age groups are warranted. ~~Our results are~~It will also ~~based on patient residential address; this may not necessarily be~~ important to understand the location differential impacts of the patient during a tropical cyclone ~~on health outcomes by geography, as well as socio-economic and demographic factors.~~

Page 5: Could this peak be capturing substituted hospitalization for care missed due to the disaster, such as home health, in patient rehab, MD offices closed so only place for care? The decline in cancer hospitalizations would seem to be related to missed appointments – some cancelled by patients and others by providers. There are several papers that address this (Radcliff et al. using VA data)

We agree with the Reviewer that some of the peaks evident in Figure 3 may be in part due to care missed, including ambulatory care. In several places in the Discussion, we highlight this possibility (for example, P. 8, Lines 186-188; P. 9, Lines 209-212; P. 10, Lines 217-219; PP. 12-13, Lines 286-288; P. 13, Lines 298-300). We have also added explicit examples of other kinds of care potentially missed due to the disaster, including ambulatory care (P. 9, Lines 196-201):

*Cancelled inpatient appointments might also play a key factor here, with non-emergency procedures being delayed or rescheduled.³⁹ The subsequent peak one to three days after exposure may in part be driven by patients visiting the hospital for care missed at locations other than the hospital (e.g., at home or at the family physician's offices) due to disruption from a tropical cyclone. There is also evidence that proximity to a tropical cyclone's path may result in the area's ambulatory (outpatient) care being disrupted.*³⁹

The Reviewer also highlights how the pattern shown in hospitalizations from cancers may also be due missed appointments. In the Discussion, we highlight the possibility that the reduction in hospitalizations from cancers was potentially due to several factors, including damage to infrastructure and supply line disruption appointments (PP. 12-13, Lines 286-288). We also suggest that the reduction in hospitalizations due to cancers may also be driven by the reduction in the delay of some pre-arranged admissions for patients with known chronic cancers (P. 13, Lines 290-292).

In any case, it is important to know how distribution of hospitalizations is impacted in the immediate aftermath of a tropical cyclone, since the daily disruption to hospitalizations has ramifications for hospital capacity and planning. Furthermore, we show that there are overall increases in hospitalizations for some outcomes (manuscript Figure 6), which shows that the disruption is not merely the moving around of patient visits, but an actual overall increase in hospitalizations, particularly for respiratory diseases and injuries. Future large-scale studies can build upon how and where the hospitalizations happened with more detailed data, such as cases in nursing homes. Relevant to this, we have added a line in the Discussion of the revised manuscript (P. 16, Lines 367-371):

Further work is needed to specifically understand which hospitals would need to be prepared with the forecast of a tropical cyclone. ~~Our inpatient records only capture emergency room visits if they become inpatient visits; records of those who are dismissed after an emergency visit are important to understand hospital capacity needs., along with which sources of health care are disrupted.~~

In addition, we have performed an extra analysis to understand the differences in association of tropical cyclones with emergency and non-emergency hospitalizations, which is shown in the Figure below, now in the revised Supplementary Information as Supplementary Figure 1. As the Figure illustrates, there are some distinct differences in how emergency hospitalization rates change after tropical cyclone exposure compared to non-emergency hospitalizations. For instance, cancer non-emergency hospitalization rates decrease at first. In contrast, cancer emergency hospitalization rates show no change on the day after to the end of the lag period (days 1 through 7), though there is still a decrease on the day of the tropical cyclone exposure. This is in agreement with the Reviewer's comments that missed or cancelled appointments may be driving the reduction in hospitalization rates for cancers after tropical cyclone exposure.

We have summarized the findings of this new analysis in the revised manuscript (PP. 5-6, Lines 111-123):

For several causes (cardiovascular diseases, endocrine disorders, genitourinary diseases, infectious and parasitic diseases, nervous system diseases, and skin and subcutaneous tissue diseases) hospitalization risk followed a similar pattern, decreasing on the day of exposure, peaking one to three days later, and gradually returning to the rate expected during unexposed days within about a week. We also examined the association between tropical cyclone exposure and daily hospitalization rates by type of hospital admission (emergency vs. non-emergency; Supplementary Figure 1). Generally, non-emergency hospitalization rates decreased in the first few days after tropical cyclone exposure before returning to no change in subsequent days, with the exception of infectious and parasitic diseases, for which we estimated increases in non-emergency hospitalizations for lags 1, 2, and 4. Emergency hospitalization rates for cardiovascular diseases, respiratory diseases, and injuries increased in the days after tropical cyclone exposure, with other causes generally showing lower or no decreases across days, in comparison to non-emergency hospitalizations.

We have also added a summary sentence in the Discussion in the revised manuscript (P. 8, Lines 180-182):

Changes in emergency hospitalization rates drove increases in hospitalization rates, with decreases driven by reductions in non-emergency hospitalizations.

Supplementary Figure 1: Percentage change in hospitalization rates with tropical cyclone exposure by cause of hospitalization, type of hospital admission and lag time (days after tropical cyclone exposure). Dots show the point estimates and error bars represent Bonferroni-corrected 95% confidence intervals.

Page 5: What does a 100% increase mean? What are the raw numbers? Is there a way to report rates as part of the numbers in Lines 127-134?

All of our results in manuscript Figures 3, 4, and 5 are given as relative (percentage) changes in hospitalization rates, which we describe in the Methods section of the original manuscript. The coefficient estimated from the statistical model is the log rate ratio between exposed and unexposed, accounting for the matching structure and adjusting for confounders. A 100% increase, therefore, indicates a doubling in the rate of hospitalizations per hurricane exposure. For more clarity, we have added an explicit explanation of what the percentages mean before describing those Figures in the Results section (P. 5, Lines 104-105; P. 6, Lines 125-127; PP. 6-7, Lines 138-144):

We present these results in Figure 3, which displays results as relative (percentage) changes in hospitalization rates after tropical cyclone exposure.

In Figure 4, we present average relative (percentage) changes in hospitalization rates across the eight examined lag days across the 13 categories in the main analysis, as well as for sub-categories with at least 50,000 hospitalizations during our study period.

In addition, we examined the distinct impact of tropical cyclone exposures in which the county's peak sustained wind was hurricane force (Beaufort scale hurricane-force winds, ≥ 64 knots) compared to tropical cyclone exposures with lower local winds (Beaufort scale gale- to

violent storm-force winds, ≥ 34 to < 64 knots) (Figure 5). Of the 2,547 county-day exposures (Figure 1), 116 (5%) were hurricane force and came from 20 hurricanes. Across categories, the relative (percentage) changes in hospitalization rates during hurricane-force exposures broadly amplified the overall tropical cyclone effects presented in Figure 3.

The numbers quoted from Figure 6, on which the Reviewer has asked for clarification, are indeed based on the relative change in hospitalization rates from the 13 main causes of hospitalization that populate Figures 4 and 5. We have added some additional clarification in describing Figure 6 to clarify this (P. 7, Lines 149-154):

Finally, we estimated the total number of additional hospitalizations for tropical cyclone exposure per decade in the week following the day of exposure across all counties included in our analysis. We used the ~~resultant risk~~ relative (percentage) changes in hospitalization rate estimates of each category on day of and each day after exposure, as shown in Figures 3 and 4, along with the average hospitalization rates during May to October in 1999 – 2014 and average decadal tropical cyclone exposure, described in detail in Methods.

Discussion: People may be deterred from seeking care if evacuation orders are issued, if EMS services are suspended for non-evacuees. There is an issue with just identifying health outcomes with no real plausible relationship with intensity or duration of disaster exposure (e.g., leukemia).

We agree with the Reviewer and have added an explicit mention about how evacuation orders may act as a deterrent to those seeking care, in addition to the other reasons that behavior may change in the week after tropical cyclone exposure (P. 9, Lines 203-206):

Tropical cyclone wind exposure can impact hospitalizations via direct (e.g., from physical trauma during exposure) or indirect (e.g., disrupting normal care management at local health care providers, causing damage to critical infrastructure which subsequently impacts health or via longer-term impacts from stress) pathways.⁶

We also agree with the Reviewer that lack of biological plausibility for the observed relationships may be of concern. However, ours is the first study to comprehensively examine the association between tropical cyclones and hospitalizations, providing a framework to analyze many different main causes and sub-causes of hospitalizations. And though there may not be well-defined biological pathways, behavioral and psychological impacts are also important. Our findings, therefore, can inform future studies to investigate specific outcomes in more detail. We have added to the existing text in the Discussion of the revised manuscript (P. 16, Lines 373-378):

There is some limited evidence to suggest that there are measurable long-term impacts on health in the years after a disaster.^{34,35} ~~Characterizing longer-term health impacts of tropical eyelones is critical~~^{47,48} There are plausible causal links between health outcomes and tropical cyclone exposure for many of the associations here,^{6,28-33} but more work needs to be done to identify and formalize these pathways.

Page 8: There are most definitely cases where disaster exposure prevents normal care management. Please see POST KATRINA

We agree with the Reviewer and have added an explicit mention of disruption to normal care management to the list of indirect impacts from tropical cyclone exposure (P. 9, Lines 203-206):

Tropical cyclone wind exposure can impact hospitalizations via direct (e.g., from physical trauma during exposure) or indirect (e.g., disrupting normal care management at local health care providers, causing damage to critical infrastructure which subsequently impacts health or via longer-term impacts from stress) pathways.⁶

Page 9: There are data on most frequent injuries and time periods in which they are most likely.

We have now reworked this section to explicitly mention the kinds of injuries which are likely during the immediate period after a tropical cyclone (P. 10, Lines 223-230):

~~Injury hospitalizations may be impacted by tropical cyclone exposure both directly and indirectly. Direct causes of injury during tropical cyclone exposure include blunt trauma, puncture wounds, lacerations, and others.⁶are impacted by tropical cyclone exposure both directly and indirectly. During and immediately after tropical cyclones exposure, common injuries originate from transport accidents, structural collapse of buildings, wind-borne debris, falling trees, and downed power lines.³² Days after exposure, other injuries such as puncture wounds, lacerations, falls from roof structures, chainsaw mishaps, and burns take more prominence in hospitalizations.³²~~

Page 9: The multiple mentions of supply lines for essential medications seems a stretch. In 1 or 2 days? Tropical storms have highly improved forecasting and warning time in days, not hours or minutes. Supplies can be stockpiled and supply lines and logistics systems fortified in advance.

We recognize that supply lines for essential medications to patients would likely not be disrupted due to depletion of stock at the local supply source, such as pharmacies. Rather, it is likely that damage to transport infrastructure makes it difficult for residents in exposed counties to access the local supply source or that the pharmacy itself is closed. Our revised manuscript clarifies this point (P. 11, Lines 250-259):

Tropical cyclone exposure could also indirectly lead to increases in acute cardiovascular disease hospitalizations, due to increased stress and physical challenges brought about during and following exposure,⁶ ~~as well as disrupting supply lines for Disruption of access to essential medicines from closure of local supply sources, such as pharmacies, may also contribute to negative cardiovascular health outcomes.^{45,46} Although we did not observe short-term changes (i.e.,⁶Although we did not observe short-term changes (i.e.,~~ in first week after tropical cyclone exposure) in cardiovascular disease hospitalizations, longer-term negative impacts of tropical cyclone exposure on cardiovascular diseases have been observed, several years after exposure itself.³⁴⁴⁷

Page 10: Stress is the result of far more than property damage or loss of life. Financial concerns, intimate partner violence, insecurity of all types is exacerbated by a disaster's impacts.

We thank the Reviewer for bringing these to our attention. The revised manuscript mentions more sources of stress which are likely to materialize during the immediate period after a tropical cyclone (P. 12, Lines 280-282):

Tropical cyclone exposure can cause stress and anxiety following potential financial concerns, intimate partner violence, loss of property, loss of family and friends, and other sources of insecurity.^{28,40,49}

Page 11: Line 245: Missed dialysis should be mentioned.

We thank the Reviewer for this suggestion. We have added this to the revised manuscript (P. 13, Lines 302-305):

Access to dialysis due to renal failure in the aftermath of a tropical cyclone would also rely on constant supply of electricity, which—when cut at home,—or unavailable at a local care provider—may result in additional hospitalizations for fluid and electrolyte disorders.^{38,51}

It should be mentioned that the number of infectious diseases are relatively small. Again, shelter surveillance for respiratory and GI conditions to address possible outbreaks.

We thank the Reviewer for this suggestion. We have mentioned that the overall number of infectious diseases is small (P. 14, Lines 326-329):

When a storm or hurricane passes through, stagnant and unclean water is often left behind,^{52,53} which can be optimal breeding grounds for many diseases, including infectious and parasitic, skin and subcutaneous, blood and digestive system diseases,^{53,54} though absolute numbers are small compared with other hospitalization causes.

Line 257: No need to bring COVID to this paper. Start this paragraph with Disasters, such as...the other statements are an overreach.

We have removed any reference to COVID-19.

Line 261: Health care systems have many provisions in place such as shelter in place to ensure personnel are in place.

We have revised the manuscript to incorporate this comment (P. 15, Lines 339-343):

While some cause-specific hospitalization rates may not change on average in the week following a tropical cyclone, the distribution of hospitalization rate changes during the post-cyclone week requires careful planning. Although many health care systems already have provisions in place, findings from our study may further inform planning.

Page 12, Line 283: This study doesn't address capacity. In natural disasters, hospital emergency capacity is also another literature. Should not be mentioned here. Again, the number of deaths directly caused by disasters in the U.S. is relatively small.

We have removed the sentence from the revised manuscript.

Page 13: We have highly improved and relatively accurate forecasting. Telemedicine and remote prescription is again an over reach. Not at all clear how this is relevant to hospitalization during tropical cyclones.

We have removed references to telemedicine and remote prescription from the revised manuscript.

Reviewer 2

Parks and colleagues leveraged a large Medicare dataset in the US with 70 million hospitalizations over 16 years to examine how tropical cyclone wind exposures affect hospitalizations from 13 disease categories and common sub-categories. They found that tropical cyclone exposure was associated with increases in hospitalizations from respiratory diseases, infectious and parasitic diseases, and injuries, but decreased cancer-related hospitalizations and no consistent changes in cardiovascular disease hospitalizations. This study is by far the most comprehensive study investigating the effects of tropical cyclone exposures on cause-specific hospitalizations. The study is well-conducted in terms of both exposure assessment and statistical methods. I enjoyed reading this manuscript. The discussions on both biological plausibility and limitations are remarkably clear and well acknowledged. Thus, I only have a few minor comments for the authors to consider in improving this paper.

We thank the Reviewer for this thoughtful and generous assessment. We have responded point-by-point to the Reviewer's questions and comments below.

Specific comments

1. Is it possible to compare the difference in hurricane-related risks between emergency room visits and other hospitalizations? Given the decreases in hospitalizations from cancer and some specific cardiovascular diseases, and the authors' argument that chronic and "non-emergency procedures being delayed or rescheduled," it would be interesting to see if the hurricane exposures would increase cancer or cardiovascular hospitalizations transferred from emergency room visits.

We agree with the Reviewer that this would be an interesting analysis, and did additional work to understand the differences in association of tropical cyclones with emergency and non-emergency hospitalizations. This is shown in the Figure below, now in the revised Supplementary Information as Supplementary Figure 1. Our additional analysis reinforces our suggestion that non-emergency appointment delays or rescheduling are likely driving the reduction in hospitalization rates in the first few days after tropical cyclone exposure for several causes, including cancer and cardiovascular disease hospitalizations. The Figure below also shows an increase in emergency cardiovascular disease hospitalizations rates 2-7 days after tropical cyclone exposure, in contrast to non-emergency hospitalizations, which decrease or remain the same. Cancer emergency hospitalization rates remain unchanged, apart from the day of tropical cyclone exposure, whereas non-emergency hospitalization rates remain lower for 0-2 days after tropical cyclone exposure.

We have summarized the findings of this new analysis in the revised manuscript (PP. 5-6, Lines 111-123):

For several causes (cardiovascular diseases, endocrine disorders, genitourinary diseases, infectious and parasitic diseases, nervous system diseases, and skin and subcutaneous tissue diseases) hospitalization risk followed a similar pattern, decreasing on the day of exposure, peaking one to three days later, and gradually returning to the rate expected during unexposed days within about a week. We also examined the association between tropical cyclone exposure and daily hospitalization rates by type of hospital admission (emergency vs. non-emergency; Supplementary Figure 1). Generally, non-emergency hospitalization rates decreased in the first few days after tropical cyclone exposure before returning to no change in subsequent days, with the exception of infectious and parasitic diseases, for which we estimated increases in non-emergency hospitalizations for lags 1, 2, and 4. Emergency hospitalization rates for cardiovascular diseases, respiratory diseases, and injuries increased in the days after tropical cyclone exposure, with other causes generally showing lower or no decreases across days, in comparison to non-emergency hospitalizations.

We have also added a summary sentence in the Discussion in the revised manuscript (P. 8, Lines 180-182):

Changes in emergency hospitalization rates drove increases in hospitalization rates, with decreases driven by reductions in non-emergency hospitalizations.

Supplementary Figure 1: Percentage change in hospitalization rates with tropical cyclone exposure by cause of hospitalization, type of hospital admission and lag time (days after tropical cyclone exposure). Dots show the point estimates and error bars represent Bonferroni-corrected 95% confidence intervals.

2. Please briefly justify the reason for choosing wind speed as the exposure metric for tropical cyclones, but not other metrics such as rainfall or flooding. Are the wind field modelling data validated against observations from weather stations?

We largely based our analysis on understanding the association between tropical cyclones and hospitalizations in line with the definition for emergency preparedness for disasters. When a storm's maximal sustained winds reach 34 knots (62 km/h, 17 mph; gale-force wind on the Beaufort scale), it is defined as a tropical cyclone and if it exceeds 64 knots (118 km/h, 73 mph; violent storm-force wind on the Beaufort scale) as a hurricane. The World Meteorological Organization (WMO) gives names to tropical cyclones based on their tropical storm-level winds,(WMO, 2020) which the National Oceanic and Atmospheric Administration (NOAA) then uses to plan forecasts and emergency planning. In addition, a smaller-scale study examined how different cyclone-related characteristics relate to respiratory disease hospitalizations and found that wind speed was the most strongly-associated with outcomes.(Yan et al., 2020)

In our study, we used wind speed as a surrogate for an overall tropical cyclone exposure. However, tropical cyclones are multi-hazard events, also characterized by other features, such as rainfall and flooding. To explicitly investigate associations with distinct tropical cyclone-related hazards, a multi-hazard analytical framework would be required and is outside the scope of this study. Therefore, to provide a parsimonious study of health impacts of tropical cyclones with as much relevance as possible to existing frameworks of disaster preparedness, we decided to focus on wind-based definitions of tropical cyclones. We have discussed this in the revised manuscript (P. 16, Lines 378-380):

We also focused on defining a tropical cyclone by wind speed, as it has direct relevance for identifying a tropical cyclone and therefore emergency planning.^{33,61}

The differential impact of tropical cyclones based on their exposure profiles is a potential area of research that may aid forecasting hospitalization burden more accurately. We have therefore added to the revised manuscript (P. 16, Lines 380-383):

Understanding in more detail whether including more information about specific tropical cyclone-related hazards, such as rainfall and flooding, in combination with wind, modify the impact of tropical cyclones on health outcomes will be an important direction of future research.

The wind field data are validated against observations from weather stations. We have added this detail to the revised manuscript (P. 26, Lines 638-640):

*In brief, an exhaustive assessment of tropical cyclones was generated from those recorded in the HURDAT2 dataset based on wind field modeling and validation against observations from weather stations.*⁶⁴

3. In the primary statistical model, why not control for potential confounding from relative humidity?

Relative humidity in our study area can be impacted by a tropical cyclone via heavy rainfall and storm surges,(Khouakhi, Villarini, & Vecchi, 2017; L. Wu et al., 2012) i.e., relative humidity can succeed exposure and, thus, act as a mediator for some of the outcomes in our

study. Adjusting for potential mediators in the statistical model can explain some of the association of interest away, thus biasing effect estimates towards the null.(Rothman, Greenland, & Associate, 2014) Therefore, we have not adjusted for relative humidity.

4. Page 6, Lines 136-137. Please briefly describe the sensitivity analyses and associated findings here.

We have added more detail from the Methods section to this part of the revised manuscript (P. 8, Lines 165-167):

We also fit models (1) including the temperature on the day of tropical cyclone exposure, as well as temperature terms of up to seven days after exposure and (2) without a temperature term. Our results were robust to these sensitivity analyses.

5. Lines 294-295. This statement is not based on the results of this study. During storms that are often leading to power outage or Internet connection lost, telemedicine might not be a good choice.

Yes, thank you! We have removed this from the revised manuscript.

Reviewer 3

Tropical cyclone exposure is associated with increased hospitalization rates

Thank you very much for the opportunity to review the manuscript entitled, Tropical cyclone exposure is associated with increased hospitalization rates, submitted for consideration for publication in Nature Communications.

This paper makes an important contribution to the field by presenting a cogent and consistent analysis of the patterning of 70 million Medicare hospitalizations over a 16-year period of analysis. An innovative approach has been devised for examining tropical cyclone exposure based on wind field modelling in order to comprehensively map counties experiencing at least one day of gale force winds associated with tropical systems during the 16-year analysis window that could then be matched to Medicare hospitalizations over the subsequent 7-day period.

This yielded an array of Gulf of Mexico and eastern seaboard counties, along coastlines but also extending far inland because these tropical systems retained their strong wind hazards, and sometimes, their cyclonic signatures, for prolonged periods as they moved over large geographic expanses. This was a brilliant decision that provides a comprehensive look at tropical cyclone impact on population health, at least in the one dimension analyzed, patterns of Medicare hospitalization. The massive size of the data sampled permitted a nuanced presentation of hospitalization patterns that rationally fit the realities of storm impacts over multiple hurricane seasons.

The series of figures showcases the results in a vivid manner that allows the reader/viewer to easily understand the patterns of hospitalization by disease category and days post-impact. This is an elegant portrayal of the data.

We thank the Reviewer for this thoughtful and generous assessment. We have responded point-by-point to the Reviewer's questions and comments below.

Recommendations for consideration:

Please consider specifying in the title that the focus is on Medicare hospitalizations. This represents a very important subset of inpatient admissions. This is an older subpopulation with much higher utilization rates and prevalent morbidities and comorbidities. Tropical cyclones alter their utilization patterns as clearly depicted. Nevertheless, patterns of injury and health services utilization for storm-related conditions are likely to be quite different for younger residents in the same storm-affected areas. The specificity of this paper, focusing on the Medicare population, is clarified right away in the abstract but should also be conveyed in the title.

We thank the Reviewer for the suggestion. We have amended the title to ‘Tropical cyclone exposure is associated with increased hospitalization rates in older adults’.

There is a cursory mention of anthropogenic climate change in the introduction along with a cluster of citations. Several important papers, including several by Kossin might wisely be added. Also, the Knutson paper (#15) in BAMS is actually Part II of a 2-part series. Consider citing Part I also, as it is equally or more relevant.

We agree and have added these references to the revised manuscript (P. 2, Lines 35-37):

The intensity of tropical cyclones is predicted to change, ~~along with their impacts on health, due to anthropogenic climate change.~~¹⁰⁻¹⁵¹⁰⁻¹⁹ Land subsidence²⁰ and increases in the proportion of impervious surfaces²¹ may further exacerbate cyclone impacts.

It might be wise to add or expand on two points in the discussion. Please consider:

First, please explain why were analyses presented only through 2014? Some of the most remarkable and active storm seasons have occurred from 2015 through 2020. These were the years that included such memorable storms as Matthew, Harvey, Irma, Maria, Florence, Michael, and Dorian.

We absolutely agree with the Reviewer that including data from more recent hurricane seasons would have given us the opportunity to include in analyses these memorable storms. However, this would require extensive data harmonization and would greatly delay this publication, which we think is very timely, especially after the catastrophic record-breaking 2020 season. Specifically, to identify cause-specific hospitalizations, we used the International Classification of Diseases (ICD) coding system. In the United States, the ninth revision, clinical modification (ICD-9-CM) was used until 2014. After this, in 2015, the United States switched from the ninth revision to tenth revision. Therefore, the data for our study period (1999 – 2014) use a consistent system for the assignment of medical cause of hospitalization. This change in coding system in the United States has caused issues arising in emergency departments from clinically-incorrect coding of hospitalizations.(Krive et al., 2015) Extending the study period past 2014 would potentially cause considerable coding issues for many causes of hospitalizations and would have led to large jumps in hospitalization rates, which would impact the output of the model reliability.

We have added comments in the revised manuscript in the Discussion encouraging future studies to include more recent years (P. 16, Lines 383-386):

Our study included millions of hospitalizations over a decade across all counties impacted by tropical cyclones in the United States during this period. In more recent years, however, many catastrophic tropical cyclones have made landfall and future studies should include these data.

Second, especially with progressively increasing influences of climate change on tropical cyclone behavior, the water hazards (storm surge, coastal wave action, extreme rainfall totals and precipitation rates, widespread inland freshwater flooding) are equally prominent hazards that both increase demand for hospitalization but also impede access. Take the example of Hurricane Harvey in 2017 (outside your window of analysis but well known). Winds quickly died down to below tropical storm force but with the center of circulation remaining near-stationary over the western Gulf, the storm unleashed 33 trillion gallons of rain on Texas and a portion of Louisiana over five days (with rainfall totals over 60 inches in some areas). Same for Florence over the Carolinas in 2018. This might be worth a mention. If data sets are available, this might make for an intriguing future analysis.

We agree with the Reviewer, and Reviewer 2 who also discussed other hazards, that this would certainly make for an interesting future analysis. We have added comments to the revised manuscript in the Discussion relevant to this (P. 16, Lines 380-383):

Understanding in more detail whether including more information about tropical cyclone-related hazards, such as rainfall and flooding, in combination with wind, modify the impact of tropical cyclones on health outcomes will be an important direction of future research.

Authors are wise to indicate that future analyses should be extended to persons who are younger than the Medicare population.

Thank you!

Because of the focus on inpatient hospitalization and the decision to limit the timeframe to seven days post-impact, neuro-psychiatric hospitalizations show minimal changes except for a blip of delirium and dementia cases. Nonetheless, outpatient and inpatient consultation for cyclone-associated PTSD and mood disorders are likely to become an important feature of health care utilization starting 5-6 weeks following impact. Optional, but might be worth a comment when you discuss this cluster of diagnoses.

We absolutely agree with the Reviewer that is critical to also examine different risk periods, including longer than a week post exposure. Our group is very interested in the impact of tropical cyclones on chronic outcomes and such analyses are already under way. Our aim for this study was to comprehensively characterize the relationship between tropical cyclones and numerous cause-specific hospitalizations during the week following exposure. We have, therefore, very carefully selected a study design and an analytical framework that are most appropriate for investigation of acute effects. Investigations of longer-term health impacts of tropical cyclones would require etiological hypotheses and modelling structures distinct from the scope of this study.

We encourage more studies focusing on longer-term outcomes in the Discussion (P. 16, Lines 375-378):

There are plausible causal links between health outcomes and tropical cyclone exposure for many of the associations here,^{6,28-33} but more work needs to be done to identify and formalize these pathways. Characterizing longer-term health impacts of tropical cyclones is critical.

Finally, the tone of your commentary about what you are observing for each disease category “sounds” a bit speculative and uncertain as I read what is written. For example, you speculate about the dramatic surge in respiratory admissions for disorders like COPD (CLRD) and mention a bit hesitantly that perhaps this could be due to power outages. This is almost certainly the case. As one example, the frequently-hit state of Florida has a designated special needs shelter (“SpNS”) in every county and a list of persons registered for pick up and transport to this retrofitted facility that is equipped with a massive, elevated (above flood waters) auxiliary generator that maintains power for days when local electrical service in the area is completely offline. Inside, these electrically-dependent (typically oxygen-dependent) local residents are able to plug in their concentrators and maintain the life-giving flow of oxygen even as the storm is raging outside (this reviewer has been onboard a SpNS during multiple hurricanes and the system works well). I add this lengthy comment to say, the one voice that seemed to be missing was that of someone who has considerable direct, onsite storm experience to provide a more confident tone to your interpretations.

We agree with the Reviewer and have rephrased sections of the Discussion in the revised manuscript to reflect more certainty.

This is an excellent paper and an important addition. Thank you for dedicating your time to preparing and clearly presenting your analysis.

We thank the Reviewer for the kind and thoughtful words above.

References

- Anderson, G. B., Ferreri, J., Al-Hamdan, M., Crosson, W., Schumacher, A., Guikema, S., ... Peng, R. D. (2020). Assessing United States county-level exposure for research on tropical cyclones and human health (in press). *Environmental Health Perspectives*.
- Armstrong, B., Gasparrini, A., & Tobias, A. (2014). Conditional Poisson models: a flexible alternative to conditional logistic case cross-over analysis. *BMC Medical Research Methodology*. <https://doi.org/10.1186/1471-2288-14-122>
- Baggett, J. (2006). Florida disasters and chronic disease conditions. *Preventing Chronic Disease*.
- Bobb, J. F., Ho, K. K. L. L., Yeh, R. W., Harrington, L., Zai, A., Liao, K. P., & Dominici, F. (2017). Time-course of cause-specific hospital admissions during snowstorms: an analysis of electronic medical records from major hospitals in Boston, Massachusetts. *American Journal of Epidemiology*, *185*(4), 283–294. <https://doi.org/10.1093/aje/kww219>
- Bobb, J. F., Obermeyer, Z., Wang, Y., & Dominici, F. (2014). Cause-specific risk of hospital admission related to extreme heat in older adults. *JAMA - Journal of the American Medical Association*. <https://doi.org/10.1001/jama.2014.15715>
- Bourque, L. B., Siegel, J. M., Kano, M., & Wood, M. M. (2006). Weathering the storm: The impact of hurricanes on physical and mental health. *The ANNALS of the American Academy of Political and Social Science*. <https://doi.org/10.1177/0002716205284920>
- Burton, L. C., Skinner, E. A., Uscher-Pines, L., Lieberman, R., Left, B., Clark, R., ... Weiner, J. P. (2009). Health of Medicare Advantage plan enrollees at 1 year after Hurricane Katrina. *American Journal of Managed Care*, *15*(1), 13–22.
- Carroll, R. J., Ruppert, D., Stefanski, L. A., & Crainiceanu, C. M. (2006). *Measurement error in nonlinear models: A modern perspective, second edition. Measurement Error in Nonlinear Models: A Modern Perspective, Second Edition*.
- CDC Healthy Aging Program. (2007). *CDC's Disaster Planning Goal: Protect Vulnerable Older Adults*. Retrieved from https://www.cdc.gov/aging/pdf/disaster_planning_goal.pdf
- Diaz, J. H. (2004). The public health impact of hurricanes and major flooding. *The Journal of the Louisiana State Medical Society : Official Organ of the Louisiana State Medical Society*.
- Furukawa, K., Otsuki, M., Kodama, M., & Arai, H. (2012). Exacerbation of dementia after the earthquake and tsunami in Japan. *Journal of Neurology*. <https://doi.org/10.1007/s00415-011-6329-x>
- Galloway, D., Jones, D. R., & Ingebritsen, S. E. (2000). Land subsidence in the United States. *US Geological Survey Circular*, (1182), 1–175.
- Greenough, G., McGeehin, M., Bernard, S. M., Trtanj, J., Riad, J., & Engelberg, D. (2001). The potential impacts of climate variability and change on health impacts of extreme weather events in the United States. *Environmental Health Perspectives*. <https://doi.org/10.2307/3435009>
- Haines, A., & Ebi, K. (2019). The imperative for climate action to protect health. *New England Journal of Medicine*, (380), 263–273.
- Hart, J. E., Liao, X., Hong, B., Puett, R. C., Yanosky, J. D., Suh, H., ... Laden, F. (2015). The association of long-term exposure to PM_{2.5} on all-cause mortality in the Nurses' Health Study and the impact of measurement-error correction. *Environmental Health: A Global Access Science Source*, *14*(1). <https://doi.org/10.1186/s12940-015-0027-6>
- Hendrickson, L. A., Vogt, R. L., Goebert, D., & Pon, E. (1997). Morbidity on Kauai before and after Hurricane Iniki. *Preventive Medicine*. <https://doi.org/10.1006/pmed.1997.0196>
- Hikichi, H., Aida, J., Kondo, K., Tsuboya, T., Matsuyama, Y., Subramanian, S. V., &

- Kawachi, I. (2016). Increased risk of dementia in the aftermath of the 2011 Great East Japan Earthquake and Tsunami. *Proceedings of the National Academy of Sciences of the United States of America*. <https://doi.org/10.1073/pnas.1607793113>
- IPCC. (2013). *Climate change 2013: the physical science basis. Working group I contribution to the fifth assessment report of the intergovernmental panel on climate change*. Cambridge, UK and New York, USA: Cambridge University Press.
- Jiao, Z., Kakoulides, S. V., Moscona, J., Whittier, J., Srivastav, S., Delafontaine, P., & Irimpen, A. (2012). Effect of Hurricane Katrina on incidence of acute myocardial infarction in New Orleans three years after the storm. *American Journal of Cardiology*. <https://doi.org/10.1016/j.amjcard.2011.09.045>
- Kelman, J., Finne, K., Bogdanov, A., Worrall, C., Margolis, G., Rising, K., ... Lurie, N. (2015). Dialysis care and death following Hurricane Sandy. *American Journal of Kidney Diseases*. <https://doi.org/10.1053/j.ajkd.2014.07.005>
- Khouakhi, A., Villarini, G., & Vecchi, G. A. (2017). Contribution of tropical cyclones to rainfall at the global scale. *Journal of Climate*, *30*(1), 359–372. <https://doi.org/10.1175/JCLI-D-16-0298.1>
- Kioumourtzoglou, M. A., Spiegelman, D., Szpiro, A. A., Sheppard, L., Kaufman, J. D., Yanosky, J. D., ... Suh, H. (2014). Exposure measurement error in PM_{2.5} health effects studies: A pooled analysis of eight personal exposure validation studies. *Environmental Health: A Global Access Science Source*, *13*(1). <https://doi.org/10.1186/1476-069X-13-2>
- Knutson, T., Camargo, S. J., Chan, J. C. L., Emanuel, K., Ho, C. H., Kossin, J., ... Wu, L. (2019). Tropical cyclones and climate change assessment Part I: Detection and attribution. *Bulletin of the American Meteorological Society*, *100*(10), 1987–2007. <https://doi.org/10.1175/BAMS-D-18-0189.1>
- Knutson, T., Camargo, S. J., Chan, J. C. L., Emanuel, K., Ho, C. H., Kossin, J., ... Wu, L. (2020). Tropical cyclones and climate change assessment part II: Projected response to anthropogenic warming. *Bulletin of the American Meteorological Society*. <https://doi.org/10.1175/BAMS-D-18-0194.1>
- Kossin, J. P. (2018). A global slowdown of tropical-cyclone translation speed. *Nature*, *558*(7708), 104–107. <https://doi.org/10.1038/s41586-018-0158-3>
- Kossin, J. P., Emanuel, K. A., & Vecchi, G. A. (2014). The poleward migration of the location of tropical cyclone maximum intensity. *Nature*, *509*(7500), 349–352. <https://doi.org/10.1038/nature13278>
- Kossin, J. P., Olander, T. L., & Knapp, K. R. (2013). Trend analysis with a new global record of tropical cyclone intensity. *Journal of Climate*, *26*(24), 9960–9976. <https://doi.org/10.1175/JCLI-D-13-00262.1>
- Krive, J., Patel, M., Gehm, L., Mackey, M., Kulstad, E., Li, J. J. ‘John,’ ... Boyd, A. D. (2015). The complexity and challenges of the ICD-9-CM to ICD-10-CM transition in emergency departments. *The American Journal of Emergency Medicine*, *33*(5), 713–718. Retrieved from <http://www.ncbi.nlm.nih.gov/pmc/articles/PMC4430372/> <http://linkinghub.elsevier.com/retrieve/pii/S0735675715001333>
- Landsea, C., Franklin, J., & Beven, J. (2014). The revised Atlantic hurricane database (HURDAT2). *The National Hurricane Center*.
- Lane, K., Charles-Guzman, K., Wheeler, K., Abid, Z., Graber, N., & Matte, T. (2013). Health effects of coastal storms and flooding in urban areas: A review and vulnerability assessment. *Journal of Environmental and Public Health*. <https://doi.org/10.1155/2013/913064>
- Ligon, B. L. (2006). Infectious diseases that pose specific challenges after natural disasters: A review. *Seminars in Pediatric Infectious Diseases*.

- <https://doi.org/10.1053/j.spid.2006.01.002>
- McMichael, A. J., Woodruff, R. E., & Hales, S. (2006). Climate change and human health: Present and future risks. *Lancet*. [https://doi.org/10.1016/S0140-6736\(06\)68079-3](https://doi.org/10.1016/S0140-6736(06)68079-3)
- Noe, R., Cohen, A. L., Lederman, E., Gould, L. H., Alsdurf, H., Vranken, P., ... Mott, J. (2007). Skin disorders among construction workers following Hurricane Katrina and Hurricane Rita: An outbreak investigation in New Orleans, Louisiana. *Archives of Dermatology*. <https://doi.org/10.1001/archderm.143.11.1393>
- Nowak, D. J., & Greenfield, E. J. (2012). Tree and impervious cover in the United States. *Landscape and Urban Planning*, *107*(1), 21–30. <https://doi.org/10.1016/j.landurbplan.2012.04.005>
- Pullen, L. C. (2018). Puerto Rico after Hurricane Maria. *American Journal of Transplantation*. <https://doi.org/10.1111/ajt.14647>
- Radcliff, T. A., Chu, K., Der-Martirosian, C., & Dobalian, A. (2018). A model for measuring ambulatory access to care recovery after disasters. *Journal of the American Board of Family Medicine*, *31*(2), 252–259. <https://doi.org/10.3122/jabfm.2018.02.170219>
- Rosenheim, N., Grabich, S., & Horney, J. A. (2018). Disaster impacts on cost and utilization of Medicare. *BMC Health Services Research*, *18*(1). <https://doi.org/10.1186/s12913-018-2900-9>
- Rothman, K. J., Greenland, S., & Associate, T. L. L. (2014). *Modern Epidemiology: 3rd Edition*. Lippincott Williams & Wilkins. <https://doi.org/10.1002/hast.292>
- Shultz, J. M., Russell, J., & Espinel, Z. (2005). Epidemiology of tropical cyclones: The dynamics of disaster, disease, and development. *Epidemiologic Reviews*. <https://doi.org/10.1093/epirev/mxi011>
- Smith, K. R., Woodward, A., Campbell-Lendrum, D., Chadee, D. D., Honda, Y., Liu, Q., ... Rocklöv, J. (2015). Human health: Impacts, adaptation, and co-benefits. In *Climate Change 2014 Impacts, Adaptation and Vulnerability: Part A: Global and Sectoral Aspects*. <https://doi.org/10.1017/CBO9781107415379.016>
- Steven Picou, J., & Hudson, K. (2010). Hurricane Katrina and mental health: A research note on Mississippi Gulf Coast residents. *Sociological Inquiry*. <https://doi.org/10.1111/j.1475-682X.2010.00345.x>
- Vecchi, G. A., Delworth, T. L., Murakami, H., Underwood, S. D., Wittenberg, A. T., Zeng, F., ... Yang, X. (2019). Tropical cyclone sensitivities to CO2 doubling: Roles of atmospheric resolution, synoptic variability and background climate changes. *Climate Dynamics*. <https://doi.org/10.1007/s00382-019-04913-y>
- Wing, C., Simon, K., & Bello-Gomez, R. A. (2018). Designing Difference in Difference Studies: Best Practices for Public Health Policy Research. *Annual Review of Public Health*. <https://doi.org/10.1146/annurev-publhealth-040617-013507>
- WMO. (2020). Tropical cyclones. Retrieved October 23, 2020, from <https://public.wmo.int/en/our-mandate/focus-areas/natural-hazards-and-disaster-risk-reduction/tropical-cyclones>
- Wu, L., Su, H., Fovell, R. G., Wang, B., Shen, J. T., Kahn, B. H., ... Jiang, J. H. (2012). Relationship of environmental relative humidity with North Atlantic tropical cyclone intensity and intensification rate. *Geophysical Research Letters*, *39*(20). <https://doi.org/10.1029/2012GL053546>
- Wu, X., Braun, D., Kioumourtzoglou, M. A., Choirat, C., Di, Q., & Dominici, F. (2019). Causal inference in the context of an error prone exposure: Air pollution and mortality. *Annals of Applied Statistics*, *13*(1), 520–547. <https://doi.org/10.1214/18-AOAS1206>
- Yan, M., Wilson, A., Dominici, F., Wang, Y., Al-Hamdan, M., Crosson, W., ... Anderson, G. B. (2020). Tropical cyclone exposures and risks of emergency Medicare hospital admission for cardiorespiratory diseases in 175 urban United States counties, 1999–2010

(in press). *Epidemiology*.

Zeger, S. L., Thomas, D., Dominici, F., Samet, J. M., Schwartz, J., Dockery, D., & Cohen, A. (2000). Exposure measurement error in time-series studies of air pollution: Concepts and consequences. *Environmental Health Perspectives*, *108*(5), 419–426.
<https://doi.org/10.1289/ehp.00108419>

Reviewers' Comments:

Reviewer #1:

Remarks to the Author:

The authors have adequately addressed the reviewer comments provided, doing an extensive revision and highlighting additional literature and limitations. This is an important contribution to the literature.

Reviewer #2:

Remarks to the Author:

The authors have fully addressed my previous comments.

Reviewer #3:

None

We thank the Editors and Reviewers for their thoughtful and constructive suggestions. We have commented in response to the Reviewers' comments, as detailed below.

Reviewer 1

The authors have adequately addressed the reviewer comments provided, doing an extensive revision and highlighting additional literature and limitations. This is an important contribution to the literature.

We thank the Reviewer for this thoughtful and generous assessment.

Reviewer 2

The authors have fully addressed my previous comments.

We thank the Reviewer for this thoughtful and generous assessment.